# WHEN DOES DIVIDE AND CONQUER WORK FOR LONG CONTEXT LLM? A NOISE DECOMPOSITION FRAMEWORK

**Zhen Xu**[1], **Shang Zhu**[2], **Jue Wang**[2], **Junlin Wang**[3],
**Ben Athiwaratkun**[2], **Chi Wang**[4], **James Zou**[2,5], **Ce Zhang**[1,2]

[1]University of Chicago, [2]Together AI, [3]Duke University,
[4]Google DeepMind, [5]Stanford University

## ABSTRACT

We investigate the challenge of applying Large Language Models (LLMs) to long texts. We propose a theoretical framework that distinguishes the failure modes of long context tasks into three categories: cross-chunk dependence (task noise), confusion that grows with context size (model noise), and the imperfect integration of partial results (aggregator noise). Under this view, we analyze when it is effective to use multi-agent chunking, i.e., dividing a lengthy sequence into smaller chunks and aggregating the processed results of each chunk. Our experiments on tasks such as retrieval, question answering, and summarization confirm both the theoretical analysis and the conditions that favor multi-agent chunking. By exploring the accelerated decay of model fidelity with input length, we also explain why, for large inputs, a weaker model configured with chunk-based processing can surpass a more advanced model like GPT4o applied in a single shot. Overall, we present a principled understanding framework and our results highlight a direct pathway to handling long contexts in LLMs with carefully managed chunking and aggregator strategies.

## 1 INTRODUCTION

Large Language Models (LLMs) have drawn significant interest from both industry and academia, thanks to their ability to handle tasks ranging from open-ended question answering to complex reasoning. As these models grow in capacity, there is an increasing demand to apply them to extended texts that may span hundreds of thousands of tokens. In principle, self-attention architectures are powerful, but their reliance on quadratic operations in sequence length can make long-context tasks computationally expensive Tay et al. (2022). Moreover, even if a model technically can process long contexts, studies have reported the quality decline of output once the input surpasses a certain length (Hsieh et al., 2024). This limitation has been attributed to phenomena such as the "lost in the middle" effect, where the model forgets or mishandles portions of the input.

Existing research has explored ways to ease these difficulties. One type of approach has focused on modifying the transformer architecture to reduce memory footprint and compute, often by, for example, shaping the attention pattern through blockwise or window-based strategies (Qiu et al., 2020; Beltagy et al., 2020), low-rank approximations (Wang et al., 2020) or routing-based approaches (Kitaev et al., 2020), seeking to prolong the feasible context length without harming accuracy too severely. Although these technical improvements often extend the maximum input size, they do not guarantee stable performance when that size becomes very large.

A more functional approach divides a large input into chunks, processes each chunk with one or more worker models, and then aggregates the outputs. Retrieval-augmented pipelines are a popular example of this idea (Lewis et al., 2020; Fan et al., 2024; Wang et al., 2024b), but they often rely on ad hoc rules for aggregating results. Their effectiveness hinges on how well global dependencies are preserved. If the aggregation step is weak, cross-chunk reasoning can be lost, which leads to

suboptimal outcomes. While these chunk-based methods can lower confusion at the worker level, the completeness of the final result is not guaranteed.

In this paper, we propose a divide and conquer framework Qian et al. (2024b); Zhang et al. (2024c); Zhou et al. (2024) that offers a more systematic view at these issues. We analyze these challenges by establishing a theoretical framework that tracks three main sources of error in long-context tasks. First, *task noise* arises from cross-chunk dependencies that cannot be handled by processing each segment alone. Second, *model noise* arises from degraded performance with increasing input length. Third, *aggregator noise* emerges when partial results are combined incorrectly, even if each chunk was handled well. We show how the balance of these factors determines when chunk-based division is advantageous, and how the design of the aggregator stage plays an important role. In particular, when a task demands only mild cross-chunk reasoning, chunking can reduce model confusion with little downside. However, when cross-chunk synergy is large, only a more advanced aggregator can preserve important global connections.

We also show that model noise can worsen at a rate that outstrips the benefit of seeing the entire input at once. Beyond a certain length threshold, splitting the text can yield better performance, even if the individual worker models are weaker than a single large model. We attribute this to the superlinear growth of the model's underlying length-induced degradation with input length. Our experiments on retrieval, QA, summarization, and other tasks confirm that this superlinear effect surfaces in practice, and demonstrate how a planner can arrange prompts to keep aggregator noise manageable. Across these long context tasks, weak models handling chunks often outperform a single-shot strong model.

**Our contributions are as follows:**

- We present a theoretical framework for modeling error terms in long-context processing. This framework helps to explain when multi-agent chunking is advantageous and when it is not.
- We provide empirical evidence of accelerated performance degradation when input length is large. We then demonstrate that splitting into smaller pieces can help mitigate these effects, except in the case where cross-chunk synergy is very high.
- We show that carefully designed prompts for both worker agents and the aggregator can stably improve final performance. This allows smaller models to surpass more advanced models on certain tasks when the input length is large.

These findings broaden our understanding of how to tackle large inputs with LLMs. By breaking down the main sources of error, our framework clarifies why chunking can help, how it can fail, and how a planner can structure prompts so that partial outputs lead to correct final answers. Through both theoretical arguments and experiments, we reveal practical guidelines for handling lengthy contexts and show that well-planned division of labor can be a strong alternative to single-shot processing with a massive context window.

## 2 RELATED WORKS

### 2.1 LONG CONTEXT LLM

Developing long context LLMs has seen increasing popularity in applications such as long document understanding. Due to the attention mechanism, transformer architecture is associated with quadratic computational and memory complexity against sequence length(Tay et al., 2022), thus expensive to train and serve in practice. To mitigate this, researchers have explored various efficient transformer architectures, such as block wise attention(Qiu et al., 2020), window attention(Beltagy et al., 2020), routing attention(Kitaev et al., 2020), etc. (Dao et al., 2022) proposed flash attention and (Liu et al., 2023) designed ring attention, both contributed to efficient training of long context LLMs. The recent progress on transformer alternatives also show promises of overcoming the computational complexity of attention mechanisms, including Mamba(Gu & Dao, 2024) and Hyena(Poli et al., 2023). Also, given existing LLMs trained on short context, many research works have been presented on length extrapolation based on positional encoding(Chen et al., 2023; Peng et al., 2024; Jin et al., 2024), and retrieval-augmented generation(Lewis et al., 2020; Wang et al., 2024b), where additional design complexity is added on positional encoding adaptation and retriever. Besides, a complementary line of work aligns LLMs for long contexts via preference/reward optimization, including LOGO (Tang et al., 2024), LongReward (Zhang et al., 2024a), and LongPO (Chen et al., 2025).

## 2.2 MULTI-AGENTS

Recent research has explored the potential of multi-agent systems utilizing large language models (LLMs) for complex task-solving. In the realm of task decomposition and planning, Qian et al. (2023) proposed a framework where multiple LLM-based agents collaboratively break down tasks into subtasks and develop execution plans. Similarly, Haji et al. (2024) introduced a hierarchical mechanism where tree of thoughts reasoners are combined with thought verifiers. Zhao et al. (2024b) breaks down and process long texts using multiple agents, enabling efficient handling of 128K token contexts through task decomposition.

Beyond task breakdown and planning, recent work focused on communication mechanisms within multiple agents. Several studies (Guo et al., 2024) have investigated various approaches to enable effective information exchange between agents. Common communication paradigms such as debate is commonly used (Du et al., 2024; Chan et al., 2024; Liang et al., 2023). Alternatively, Qian et al. (2024a) adapts a cooperative paradigm where agents work together towards a common goal. Many multi-agent works use a layered structure. Wang et al. (2024a) developed a multi-layered framework that separates LLMs into proposers and aggregators. Liang et al. (2023) encourages more divergent thinking though layered debate among debaters. However, a centralized sharing structure has also been explored by Hong et al. (2023).

## 2.3 DIVIDE AND CONQUER

Divide-and-conquer is a classic computational strategy that has recently been integrated with LLM-based multi-agent systems to handle long-context processing. Recent works such as LC-Boost Qian et al. (2024b), Chain-of-Agents (CoA) Zhang et al. (2024c), LongAgent Zhao et al. (2024a), and LLM×MapReduce Zhou et al. (2024) , adopt this paradigm by splitting long inputs into smaller chunks processed by worker agents and then merging their outputs through a manager agent. These methods improve efficiency and enable long-text reasoning without requiring large context models.

However, existing approaches lack a formal theoretical framework to analyze the interaction between task complexity, model noise, and aggregation errors, making it difficult to optimize chunking strategies. They also struggle with understanding how cross-chunk dependencies impact performance, often leading to loss of contextual coherence when aggregating local outputs. This work addresses these gaps by providing a rigorous modeling framework for analyzing noise propagation in Divide-and-conquer architectures, offering a systematic approach to optimizing multi-agent collaboration for long-context tasks.

## 3 THEORETICAL MODELING FRAMEWORK

### 3.1 PROBLEM FORMULATION: THE FIDELITY DECOMPOSITION

We model the divide-and-conquer (D&C) pipeline as an information transmission channel. Let $x$ be a long input of length $T$ split into $n$ contiguous chunks. Let $S(\hat{y}) \in (0, 1]$ represent the normalized evaluation metric for a prediction $\hat{y}$. We assume $S(\cdot) > 0$ to ensure well-defined logarithmic losses.

We define the system's score fidelity $\rho \in (0, 1]$ as the ratio of the retained score to the ideal score. To analyze the error accumulation, we utilize the following telescoping identity, which decomposes the system fidelity into a product of score ratios in three stages (explained in following sections):

$$\rho_{\text{sys}} = \frac{S(h(\hat{\mathbf{a}}))}{S(y^*)} = \underbrace{\frac{S(h^*(\mathbf{a}^*))}{S(y^*)}}_{\rho_{\text{task}}} \times \underbrace{\frac{S(h(\mathbf{a}^*))}{S(h^*(\mathbf{a}^*))}}_{\rho_{\text{agg}}} \times \underbrace{\frac{S(h(\hat{\mathbf{a}}))}{S(h(\mathbf{a}^*))}}_{\rho_{\text{model}}} . \tag{1}$$

For analytical convenience, we analyze the system in log-space by defining the fidelity loss $\mathcal{L} := -\log(\rho)$. The multiplicative nature of fidelity implies additive losses:

$$\mathcal{L}_{\text{sys}} = \mathcal{L}_{\text{task}} + \mathcal{L}_{\text{agg}} + \mathcal{L}_{\text{model}}. \tag{2}$$

We also define the total system error as $\mathcal{E}_{\text{total}} := 1 - \rho_{\text{sys}}$. In the high-fidelity regime, a first-order expansion yields $\mathcal{E}_{\text{total}} \approx (1 - \rho_{\text{task}}) + (1 - \rho_{\text{agg}}) + (1 - \rho_{\text{model}})$ (Appendix B).

## 3.2 STAGE 1: DECOMPOSITION (TASK FIDELITY)

Let $y^* = f^*(x)$ be the global ground truth. Let $\mathbf{a}^* = (a_1^*, \ldots, a_n^*)$ denote the *optimal* chunk-level artifacts constrained by a fixed schema (e.g., restricted bandwidth or token budget). Specifically, $\mathbf{a}^*$ represents the information bottleneck imposed strictly by the decomposition interface. We define an ideal aggregator $h^*$ as the optimal function in the model class $\mathcal{H}$ given these artifacts:

$$h^* = \arg \max_{h \in \mathcal{H}} S(h(\mathbf{a}^*)).$$

Task fidelity measures the information retained under these decomposition constraints:

$$\rho_{\text{task}} = \frac{S(h^*(\mathbf{a}^*))}{S(y^*)}. \tag{3}$$

If $\rho_{\text{task}} \ll 1$, the task inherently resists decomposition under the chosen schema.

## 3.3 STAGE 2: AGGREGATION (AGGREGATOR FIDELITY)

Let $h$ be the actual aggregator model used in the pipeline. We define Aggregator Fidelity by comparing the actual aggregator against the ideal aggregator on the same *perfect* inputs $\mathbf{a}^*$:

$$\rho_{\text{agg}} = \frac{S(h(\mathbf{a}^*))}{S(h^*(\mathbf{a}^*))}. \tag{4}$$

This term isolates the limitation of the aggregator model itself (e.g., failure to reason over long sequences of artifacts), assuming perfect input artifacts.

## 3.4 STAGE 3: LOCAL PROCESSING (MODEL FIDELITY)

Let $\hat{\mathbf{a}}$ denote the *actual* noisy outputs generated by worker models. We define Model Fidelity as the ratio of performance given actual noisy inputs versus perfect inputs:

$$\rho_{\text{model}} = \frac{S(h(\hat{\mathbf{a}}))}{S(h(\mathbf{a}^*))}. \tag{5}$$

We assume monotonicity in expectation: replacing noisy worker outputs with ideal artifacts does not degrade performance, ensuring $\rho_{\text{model}} \leq 1$. The term $\mathcal{L}_{\text{model}}$ quantifies the loss caused strictly by worker errors.

## 3.5 THE D&C ADVANTAGE

We now formally characterize when a system of weaker agents outperforms a single stronger agent on very long inputs.

**Proposition 3.1** (The D&C Advantage). *Let $\mathcal{L}_{\text{strong}}(T)$ be the loss of a single strong model and $\mathcal{L}_{\text{D\&C}}(T)$ be the loss of a divide-and-conquer system on input length $T$. Assume:*

1. ***Super-Linear Collapse:*** *The strong model's loss grows super-linearly with context length: $\mathcal{L}_{\text{strong}}(T) = \omega(T)$ (i.e., $\lim_{T \to \infty} \mathcal{L}_{\text{strong}}(T)/T = \infty$).*

2. ***Bounded Unit Loss:*** *The D&C system processes inputs in fixed-size chunks, and the error per chunk (and associated overhead) is bounded by a constant.*

*Then, the D&C loss accumulates linearly ($\mathcal{L}_{\text{D\&C}}(T) = O(T)$), and there exists a critical threshold $T_0$ such that for all $T > T_0$, the D&C system strictly outperforms the single strong model.*

This proposition justifies D&C strategies despite the overhead: as $T \to \infty$, the super-linear "brain fog" of a single model inevitably exceeds the linear costs of decomposition and aggregation.

## 3.6 THREE REGIMES OF ERROR

Based on the decomposition, we categorize long-context tasks into three regimes:

- **Regime 1: Trivial (Negligible Noise).** $\mathcal{L} \approx 0$. Tasks like sparse retrieval where decomposition is lossless and models are robust.

- **Regime 2: The "Silo Effect" (Task Noise Dominates).** $\mathcal{L}_{\text{task}} \gg \mathcal{L}_{\text{model}}$. The task requires global reasoning that is lost under the chunking schema. In this regime, D&C strategies saturate below the optimal performance regardless of model quality.

- **Regime 3: The "Brain Fog" (Model Noise Dominates).** $\mathcal{L}_{\text{model}} \gg \mathcal{L}_{\text{task}}$. The input length $T$ is so massive that single-agent fidelity collapses. This is the optimal regime for D&C strategies.

## 4 A SIMPLE IMPLEMENTATION OF THE FRAMEWORK

We develop a minimal three-part system as illustrated in Figure 1: a **planner** to manage how the original query is segmented and how prompts are arranged, multiple **worker agents** that each handle a different sub-chunk, and a single **manager agent** that merges all sub-results. Below we briefly introduce the worker and manager roles, followed by a more detailed description of the planner.

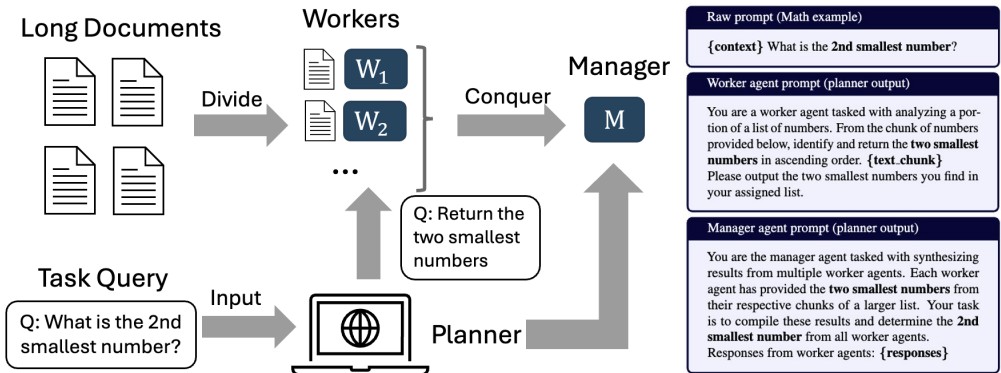

Figure 1: A simple implementation of the divide and conquer framework. The Math task example (right panel) illustrates the planner's critical role by translating instructions of returning 2nd smallest number into returning the *two smallest numbers* per chunk.

**Worker agent:** We split the input sequence into chunks of approximately equal length, and each worker agent is assigned a single chunk. Each worker focuses solely on its segment, without managing cross-segment dependencies. Although we use identical models for the workers, one can easily extend the architecture to mix different worker models as needed.

**Manager agent:** After the worker agents produce partial outputs, the manager agent merges them into one final result. In our baseline, this manager agent is the same model type as the workers, but one may employ a more specialized manager for tasks requiring deeper global reasoning.

**Planner:** The planner orchestrates how the input is divided, what instructions each worker agent receives, and how the manager agent is prompted to unify the partial outputs. Rather than having a human specify how to break down the input, we direct the planner to do so by following steps: (1) **Job Assignment.** The planner decides how many chunks to create and which segments each worker agent will process. (2) **Prompt Preparation.** Based on the task details, the planner modifies the worker prompts so that each worker's output can be correctly integrated downstream. It also sets up the manager prompt to combine partial results properly. (3) **Iterative Refinement.** The planner can run a brief evaluation step using some validation data to identify mispredicted cases. It then revises the prompt structure or chunking strategy to reduce errors. If repeated excessively on the same data, there is a risk of overfitting, so the planner typically does only a few refinements.

**Fast chunk-size estimation via sparse sampling** Selecting an appropriate chunk size is critical when model noise dominates. Motivated by our theoretical framework, we adopt a minimal-budget procedure to pick an approximately optimal chunk size without an exhaustive grid search.

**Inputs.** Candidate chunk sizes $\mathcal{C}$, a **small** per-configuration sample budget $m$, a development set $\mathcal{D}$ of tasks, and a task-specific metric M.

**Procedure.** For each $c \in \mathcal{C}$, draw $m$ random documents $S_c \subset \mathcal{D}$ (without replacement). Run the D&C pipeline with chunk size $c$ on $S_c$ and record $\hat{s}(c) = \frac{1}{m} \sum_{x \in S_c} \mathsf{M}(\mathrm{D\&C}(x; c))$. Select $c^\star = \arg\max_{c \in \mathcal{C}} \hat{s}(c)$. Deploy the D&C pipeline on the full set using $c^\star$.

**Complexity and rationale.** This reduces the search cost from $O(|\mathcal{D}| \cdot |\mathcal{C}|)$ evaluations to $O(m \cdot |\mathcal{C}|)$ with $m \ll |\mathcal{D}|$. When $\mathcal{L}_{\mathrm{model}}$ dominates and the underlying length-induced degradation $g(L)$ grows superlinearly and is near-monotone in $L$ (Sec. 3.6), the D&C error versus chunk size typically exhibits a clear near-convex optimal region; a few random samples per configuration suffice to localize this optimum.

## 5 EXPERIMENTS

### 5.1 SETTINGS

In our empirical analysis and general discussions, we use the terms task noise, model noise, and aggregator noise as intuitive, macroscopic shorthands for the performance degradations driven by their corresponding formal terms $\mathcal{L}_{\mathrm{task}}, \mathcal{L}_{\mathrm{model}}, \mathcal{L}_{\mathrm{agg}}$. We present experiments to assess how the three primary noise components—model noise, task noise, and aggregator noise—affect system performance under different context lengths and across multiple tasks. Section 3.6 indicates that the model term $\mathcal{L}_{\mathrm{model}}$ can grow more than linearly with input length, and that cross-chunk dependencies (captured by $\mathcal{L}_{\mathrm{task}}$) can impose strong requirements on any mechanism for merging partial outputs. Our experiments test the ideas on six tasks with different data scales and use several agents.

**Tasks** We experiment on six diverse tasks including: Key-Value Retrieval, Math Find Number, Summarization, Dialogue Character Inference, and Open Question QA with and without choices. These tasks are based on InfiniteBench Zhang et al. (2024b) and LongBench-V2 Bai et al. (2024) but we have modified the generation and prepared different lengths of these tasks. These tasks include: **Key-Value Retrieval (KV)**, **Math Find Number (Math)**, **Summarization (Sum)** , **Open Question QA (QA-IB and QA-LB)**, **Dialogue Character Inference (Char)**. Detailed task descriptions are provided in Appendix D.

**LLM Agents** We have experimented with diverse agents choices from commercial OpenAI models and open sourced models from Meta and Alibaba. `gpt-4o-2024-08-06` **(gpt4o)** 128K model from OpenAI. `gpt-4o-mini-2024-07-18` **(gpt4omini)** 128K model from OpenAI. `Llama-3.1-70B-Instruct` **(llama70b)** 128K 70B LLM model, `Llama-3.2-3B-Instruct` **(llama3b)** 128K 3B LLM model from Meta Grattafiori (2024). `Qwen2.5-72B-Instruct` **(qwen72b)** 32K LLM model from Alibaba QWen Yang et al. (2024).

In the experiments where divide and conquer agents are implemented, the manager and worker agents are homogeneous and the planner agent is QWen72b. The temperature is set to 0 to minimize stochasticity during decoding. In the following sections, we study these effects through three empirical lenses: single-agent length-induced degradation, task-term/decomposability effects, and aggregator effects. The robustness of these observations and the practical applicability of our framework are further underscored by comprehensive supplementary analyses (detailed in Appendices F-O), which include **utility of the framework, explorations of chunking variants, comparisons with RAG, evaluations across diverse model architectures**, and etc.

### 5.2 LENGTH-INDUCED MODEL DEGRADATION

We estimate the **single-agent** model error for input $x$ directly via the empirical task score: $\mathcal{E}_{\mathrm{single}}(x) = 1 - S(f_{\mathrm{single}}(x))$, where $S(\cdot)$ is the task-specific normalized score (e.g., Accuracy). By evaluating this for different input lengths $T = \|x\|$, we see how the model's performance declines as context grows. In Figures 2b (Math) and 2a (KV), each agent processes the entire input directly (up to 128K tokens). We measure accuracy at each length; the detailed numerical scores supporting these figures are provided in Appendix F.

For KV retrieval (Figure 2a), most agents show a downward trend in accuracy as input length increases, consistent with the Super-Linear Collapse behavior assumed in Proposition 3.1 (loss growing faster than linearly with context length). Once the length exceeds tens of thousands of tokens, performance rapidly decays.

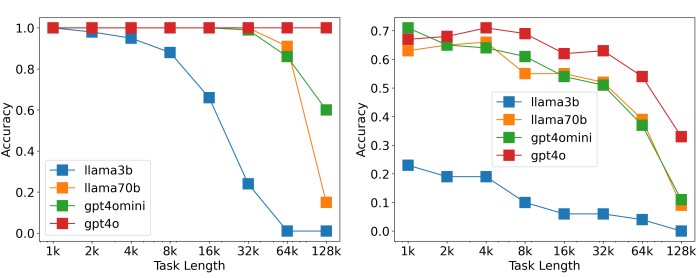

(a) Single agent model performance w.r.t input length: KV Task (b) Single agent model performance w.r.t input length: Math Task

Similar observations are seen for Math (Figure 2b), where the loss of accuracy at longer contexts is stark for gpt4omini and llama70b. Interestingly, llama3b is already too weak at shorter lengths, so its performance saturates at near-random levels. These results reinforce our theoretical statement that length-induced degradation can become the dominant error term for long $x$ (the "Brain Fog" regime in Sec. 3.6). For tasks that do not require intense synergy across the entire input, splitting the input into shorter segments (i.e., reducing $L$ per worker) should mitigate this growth, an idea formalized in Proposition 3.1.

## 5.3 TASK TERM AND DECOMPOSABILITY

Next, we examine how multi-agent performance depends on the task term (cross-chunk dependency / decomposability, $\mathcal{L}_{\text{task}}$) and the model term (length-induced degradation, $\mathcal{L}_{\text{model}}$). Since these decomposition terms are not directly observable from task metrics, we use proxies: we vary the chunk size to control per-worker context length (capturing sensitivity to length-induced model degradation), and we summarize cross-chunk dependency using a proxy based on how D&C outputs deviate from the single-agent baseline over the discrete set of chunk configurations $\mathcal{C}$.

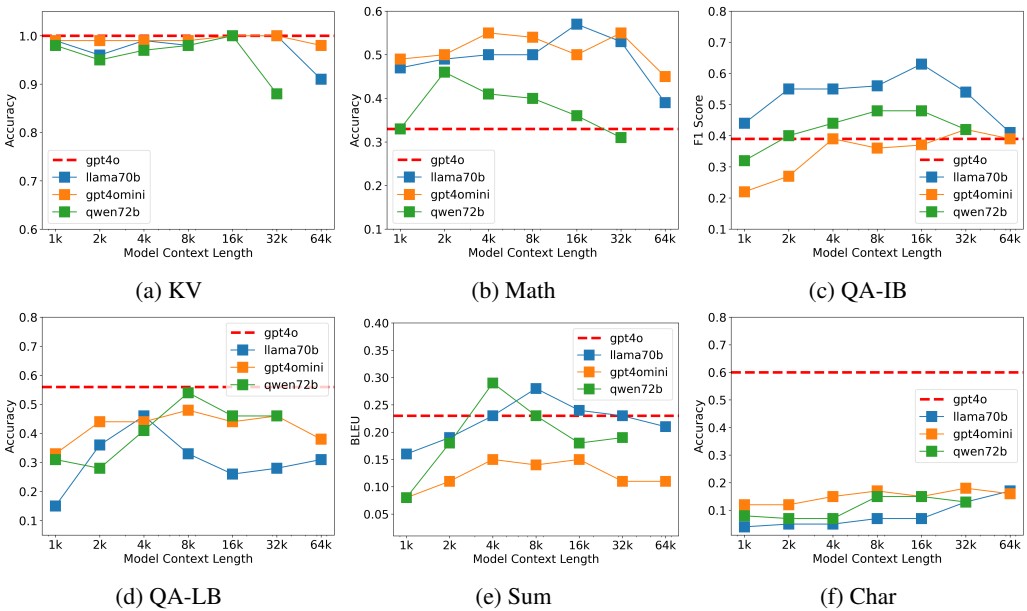

Figure 3: Joint effect of the task term (decomposability / cross-chunk dependency, $\mathcal{L}_{\text{task}}$) and the model term (length-induced degradation, $\mathcal{L}_{\text{model}}$). According to the discussion in Sec 3.6, (a) has negligible task/model terms; (b)-(e) are model-term dominated; (f) is task-term dominated.

Figure 3 illustrates the performance of five agents on the 128K version of each task, varying the chunk size from 1K to 64K tokens. The comprehensive performance metrics for all tasks and models underpinning this figure are available in Appendix G. We observe three patterns that match the

three regimes described in Section 3.6: (1) **Low task noise (KV).** When cross-chunk synergy is minimal, splitting yields similar results regardless of $n$. This matches the "low task noise, low model noise" regime, where $\mathcal{L}_{\text{task}}$ is negligible and $\mathcal{L}_{\text{model}}$ remains small, so aggregator choices have little impact. (2) **Dominating model noise (Math, QA, Sum).** In these tasks, large input length makes single-shot usage prone to confusion, so $\mathcal{L}_{\text{model}}$ becomes significant. However, $\mathcal{L}_{\text{task}}$ is moderate enough that a basic aggregator can handle partial results effectively. Splitting into smaller chunks thus reduces per-chunk confusion and outperforms a single-shot approach. This behavior aligns with Proposition 3.1, showing that the accelerated growth of model error can be contained with divide-and-conquer. (3) **Dominating task noise (Char).** In this scenario, cross-chunk interactions are so extensive that partial outputs cannot capture the global context unless the aggregator reintroduces nearly the entire input. If the aggregator does not do so, performance remains low. This corresponds to the "dominating task noise" regime, where $\mathcal{L}_{\text{task}}$ is large enough to overshadow the benefits of splitting.

We also explored whether introducing a small token overlap between chunks could mitigate task noise effects. However, as detailed in Appendix I, a 1K token overlap yielded mixed and generally marginal benefits for the Llama-70B model on 128K tasks, suggesting it does not fundamentally alter these noise trade-offs for the scenarios tested. Furthermore, to assess the generalizability of our noise framework, we evaluated additional diverse models across various tasks, as detailed in Appendix L. These results confirm that while effective context length (e.g., per Ruler benchmark) influences performance, the observed noise patterns and the utility of our decomposition are consistent, with task complexity significantly interacting with these factors across different architectures.

Overall, these empirical data show that the relative magnitudes of $\mathcal{L}_{\text{model}}$ and $\mathcal{L}_{\text{task}}$ determine whether chunking helps or hurts. Moderate-synergy tasks benefit most from a chunk-based approach, validating our theoretical insights in Section 3.6. Apart from the high-synergy character-inference problem, *divide-and-conquer methods achieve performance on par with or exceeding the strongest single-shot model*, reinforcing the idea that chunking is advantageous when the model term dominates and the task term remains modest.

## 5.4 AGGREGATOR EFFECTS

We examine the aggregator term $\mathcal{L}_{\text{agg}}$, which reflects how well the partial results are merged. Here, we compare a stronger aggregator prompt (planner-based) to a weaker one (manual) over the discrete set of chunk configurations $\mathcal{C}$. Figure 4 contrasts two approaches on the Math and QA-LB tasks for large $T$: (1) a *manual aggregator prompt*, which simply asks each worker to solve its subproblem in isolation, and (2) a *planner-based aggregator prompt*, which enforces a structured approach to partial outputs. This yields a visible performance gap (shaded area in Figure 4), showing that aggregation errors can be substantially reduced via more coordinated prompts. The same effect is seen in QA-LB, where the aggregator must reconcile partial answers from each worker. These empirical results consolidate Section 3.6 and our claim that aggregator prompting plays a major role: a robust aggregator prompt can keep $\mathcal{L}_{\text{agg}}$ modest, while naive prompts inflate the final error. The planner effectively refines aggregator instructions to align worker outputs with the manager's requirements, illustrating how a minimal overhead aggregator can preserve the theoretical advantages of chunking large inputs.

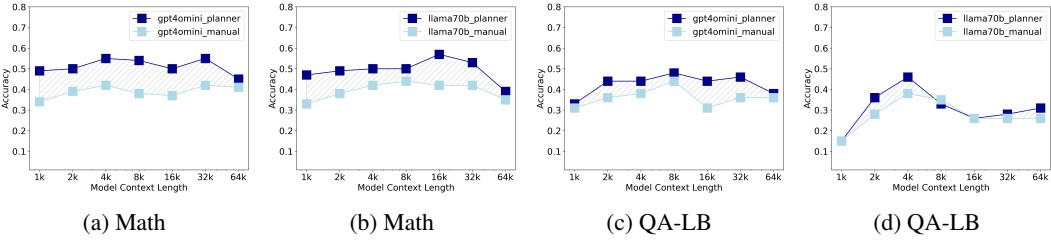

|            |            |            |            |
| :--------: | :--------: | :--------: | :--------: |
| (a) Math   | (b) Math   | (c) QA-LB  | (d) QA-LB  |

Figure 4: Aggregator errors across different tasks and models.

Table 1: Predictive Utility: Optimal Chunk Size Estimation on QA-IB and Summarization (Score (Optimal Chunk Size Found)). Total task length 128K tokens.

| Model | 3-sample | 5-sample | 10-sample | Optimal after Exhaustive Search |
|---|---|---|---|---|
| **QA-IB** | | | | |
| gpt4omini | 0.38 (64K) | 0.42 (32K) | 0.42 (32K) | 0.42 (32K) |
| llama70b | 0.55 (2K) | 0.63 (16K) | 0.63 (16K) | 0.63 (16K) |
| qwen72b | 0.40 (2K) | 0.48 (16K) | 0.48 (8K) | 0.48 (8K & 16K) |
| **Sum** | | | | |
| gpt4omini | 0.15 (16K) | 0.14 (8K) | 0.14 (8K) | 0.15 (4K & 16K) |
| llama70b | 0.23 (32K) | 0.24 (16K) | 0.28 (8K) | 0.28 (8K) |
| qwen72b | 0.23 (8K) | 0.29 (4K) | 0.29 (4K) | 0.29 (4K) |

## 5.5 FAST ESTIMATION OF THE OPTIMAL CHUNK SIZE

We evaluate the cheap-sampling estimator from Sec. 4 on tasks where model noise dominates: QA-IB and Summarization. For each candidate chunk size, we evaluate D&C on only $m \in \{3, 5, 10\}$ randomly selected documents from the 128K-token setting and pick the best-performing chunk size to deploy. Tables 1 compares this low-budget selection against the exhaustive grid search over all documents and all chunk sizes. We observe that even with three to five samples per configuration, the selected chunk sizes are near-optimal and often *exactly* match the exhaustive-search optimum, delivering the same final scores while avoiding a full grid search. This trend aligns with our framework: superlinear and near-monotone model noise in $L$ drives a clear optimal region in chunk size; a handful of samples per configuration is enough to trace the coarse error contour and locate the optimum, yielding significant computational savings without sacrificing peak performance.

## 6 CONCLUSION

In this work, we presented a theoretical and empirical framework to understand when and why the Divide-and-Conquer (D&C) strategy is effective for long-context LLMs. Our framework decomposes long-context failure modes into three fundamental fidelity loss components ($\mathcal{L}_{task}, \mathcal{L}_{model}, \mathcal{L}_{agg}$) that we intuitively refer to as task noise (cross-chunk synergy), model noise (length-induced confusion), and aggregator noise (integration flaws).

Through our formal analysis and empirical validation, we demonstrated that when the length-induced degradation (model noise) grows superlinearly with context size, chunk-based processing can effectively suppress this loss. Provided that the cross-chunk dependencies (task noise) are manageable, D&C enables a weaker model to significantly outperform a more advanced model operating in a single shot. Furthermore, we introduced an LLM-based Planner to dynamically align worker outputs, demonstrating a practical approach to minimizing aggregator errors.

Overall, by bridging the formal fidelity-loss mechanisms with observable noise patterns, our results highlight a principled and actionable pathway to handling massive contexts in LLMs through carefully managed chunking and aggregation strategies.

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

## A  USAGE OF LLMS

Large Language Models (LLMs) were minimally employed during manuscript preparation, specifically to enhance the clarity and fluency of the writing. The role of LLMs was strictly limited to minor linguistic refinement, and the authors retain full responsibility for the content and accuracy of the work.

## B  DERIVATION AND APPROXIMATIONS

This appendix details the relationship between the exact fidelity product and the additive noise approximation.

**Exact Decomposition.**  As defined in Section 3, the system fidelity is an exact telescoping product:

$$\rho_{\text{sys}} = \rho_{\text{task}} \cdot \rho_{\text{agg}} \cdot \rho_{\text{model}}.$$

By defining the log-loss $\mathcal{L} := -\log(\rho)$, the decomposition becomes exactly additive in log-space:

$$\mathcal{L}_{\text{sys}} = \mathcal{L}_{\text{task}} + \mathcal{L}_{\text{agg}} + \mathcal{L}_{\text{model}}.$$

**Linear Error Approximation.**  For high-fidelity regimes ($\rho \approx 1$), we can recover an intuitive additive error formulation. Let $\epsilon_{\text{task}} := 1 - \rho_{\text{task}}$, $\epsilon_{\text{agg}} := 1 - \rho_{\text{agg}}$, and $\epsilon_{\text{model}} := 1 - \rho_{\text{model}}$. Then

$$\rho_{\text{sys}} = \rho_{\text{task}} \rho_{\text{agg}} \rho_{\text{model}} = (1 - \epsilon_{\text{task}})(1 - \epsilon_{\text{agg}})(1 - \epsilon_{\text{model}}).$$

Expanding the product and dropping higher-order interaction terms (e.g., $\epsilon_{\text{task}}\epsilon_{\text{agg}}$) yields

$$\mathcal{E}_{\text{total}} := 1 - \rho_{\text{sys}} \approx \epsilon_{\text{task}} + \epsilon_{\text{agg}} + \epsilon_{\text{model}} = (1 - \rho_{\text{task}}) + (1 - \rho_{\text{agg}}) + (1 - \rho_{\text{model}}). \quad (6)$$

## C  JUSTIFICATION OF PROPOSITION 3.1

We analyze the crossover point where the D&C strategy outperforms a single strong agent. We derive the asymptotic behavior directly from the **Fidelity Decomposition** framework.

**1. Single Strong Agent (The Brain Fog).**  By the Super-Linear Collapse assumption, the strong model suffers from attention dispersion as context grows. Its fidelity loss grows super-linearly:

$$\mathcal{L}_{\text{strong}}(T) = \omega(T) \implies \lim_{T \to \infty} \frac{\mathcal{L}_{\text{strong}}(T)}{T} = \infty.$$

**2. Weak D&C System (Deriving Linearity).** Recall the additive decomposition of the D&C system loss:

$$\mathcal{L}_{\text{D\&C}}(T) = \mathcal{L}_{\text{task}} + \mathcal{L}_{\text{agg}} + \mathcal{L}_{\text{model}}.$$

We analyze the growth of each term under the **Bounded Unit Loss** assumption:

- **Model Loss ($\mathcal{L}_{\text{model}}$):** The input is split into $n \propto T$ chunks of fixed length. Assuming the error introduced by each local worker is bounded and independent of $T$, the accumulated log-loss scales linearly with the number of chunks: $\mathcal{L}_{\text{model}}(T) = O(n) = O(T)$.

- **Overhead ($\mathcal{L}_{\text{task}} + \mathcal{L}_{\text{agg}}$):** Assuming the task is feasible (finite $\mathcal{L}_{\text{task}}$) and the aggregation process scales linearly with the input size (or uses a hierarchical structure with bounded error), the overhead grows at most linearly: $\mathcal{L}_{\text{task}} + \mathcal{L}_{\text{agg}} = O(T)$.

Combining these terms, the total D&C loss is bounded linearly:

$$\mathcal{L}_{\text{D\&C}}(T) = O(T).$$

**3. Asymptotic Crossover.** We compare the growth rates. Since $\mathcal{L}_{\text{strong}}(T)$ grows super-linearly ($\omega(T)$) while $\mathcal{L}_{\text{D\&C}}(T)$ grows linearly ($O(T)$), the ratio diverges:

$$\lim_{T \to \infty} \frac{\mathcal{L}_{\text{strong}}(T)}{\mathcal{L}_{\text{D\&C}}(T)} = \infty.$$

This shows that there exists a threshold $T_0$ such that for all $T > T_0$, $\mathcal{L}_{\text{D\&C}} < \mathcal{L}_{\text{strong}}$.

## D    TASKS DESCRIPTION

We experiment on six diverse tasks including: Key-Value Retrieval, Math Find Number, Summarization, Dialogue Character Inference, and Open Question QA with and without choices. These tasks are based on InfiniteBench Zhang et al. (2024b) and LongBench-V2 Bai et al. (2024) but we have modified the generation and prepared different lengths of these tasks. These tasks include: **Key-Value Retrieval (KV)** We randomly generated Key-Value pairs. For a given key, the task is to retrieve the value associated. This task only evaluates the capability of retrieval. The evaluation metric is accuracy. We prepare synthetic KV task to have lengths ranging from 1K to 128K. **Math Find Number (Math)** We randomly generated a long list of integers following Gaussian distribution and the task is to find the Kth largest or smallest number in this list. Each query is slightly different in K. This task evaluates the memory and mathematical reasoning. The evaluation metric is accuracy. We prepare synthetic Math task to have lengths ranging from 1K to 128K. **Summarization (Sum)** We provide English texts and the task is to summarize them. The ground truth summarization comes from the InfiniteBench. This task evaluates the language summarization. The evaluation metric is ROUGE score. **Open Question QA (QA-IB and QA-LB)** We provide English text and some open questions for question answering. These questions are very diverse and the task requires LLM to do multi-hop reasoning to answers. The task QA-IB comes from InfiniteBench and doesn't come with choices and the task QA-LB comes from LongBench-V2 and is provided with four choices. The evaluation metrics are respectively F1 score and accuracy. **Dialogue Character Inference (Char)** We provide English dialogue scripts between many characters. One of the characters is masked and the task is to infer the name of the masked person. This task evaluates the capacity of language reasoning and will require LLM to focus on the interaction of people discussions to understand the relation. The evaluation metric is accuracy. This task comes from InfiniteBench.

## E    SAMPLE PROMPTS FOR PLANNER, MANAGER AND WORKER AGENTS

The planner provides an automated way to transform a raw prompt into structured instructions for both worker and manager agents. It parses the original task prompt, identifies key requirements, and then generates appropriate prompts for the workers and the manager. This reduces manual overhead in designing separate prompts, since the planner consistently creates focused directives for each agent.

> **Planner Prompts**
>
> Here is a prompt that I use for one LLM agent. Now since my input is too long, I divide it into multiple chunks and feed one chunk to one worker agent. Suppose now that you are the manager agent to synergize the responses of multiple worker agents in order to complete the prompt task. Write the prompt for each worker agent and the prompt for manager agent to synthesize the answer.
>
> **{RAW TASK PROMPT}**

### E.1 EXAMPLE: SUMMARIZATION

To further illustrate the planner, we use Summarization task as another example. We first show the manual (default) prompts for worker and manager agents as below.

> **Raw prompt from benchmark**
>
> Read the book paragraphs below and summarize.
> Book: **{context}**

> **Worker agent prompt (manual)**
>
> Read the book paragraphs below and summarize.
> Book: **{text_chunk}**

> **Manager agent prompt (manual)**
>
> There is a list of candidates. Each candidate has been given one sequential part of a book and summarize their own parts. You have been provided with their responses. Summarize their summaries into one.
> Responses from candidates: **{response_from_workers}**

Now we use the designed planner to automatically prepare the appropriate prompts for workers and manager. We feed the raw prompt from benchmark to the planner.

> **Worker agent prompt (planner)**
>
> You are part of a team tasked with summarizing a portion of a book. Below is an excerpt from the book assigned to you. Please provide a concise summary of this specific section, highlighting key points and themes.
> Excerpt: **{book_chunk}**
> Please ensure your summary is clear and focuses on the main ideas presented in the excerpt.

> **Manager agent prompt (planner)**
>
> You are the manager agent responsible for synthesizing the summaries provided by multiple worker agents. Each worker agent has been given a different section of the book to summarize. Your task is to combine these individual summaries into a coherent and comprehensive summary of the entire book section provided to the worker agents.
> Responses from Worker Agents: **{response_from_workers}**
> Please integrate these summaries, ensuring that the overall summary captures the main points, themes, and any important details necessary for understanding the book's content as a whole.

## E.2 EXAMPLE: QA

---

**Raw prompt from benchmark**

**{context}**
How do users most likely plan to dress for a master's graduation?
A. Wear a tuxedo or robe.
B. Dress business casual.
C. Wear business attire.
D. Wear whatever clothes they feel comfortable in.

---

**Worker agent prompt (planner)**

**{context}**
You are a worker agent. You will receive a portion of data related to how users plan to dress for a master's graduation. Your task is to analyze the provided information and offer insights based on your assigned chunk. Focus on one of the possible dress options and any relevant context or rationale behind the choice. The options include:
A. Wear a tuxedo or robe,
B. Dress business casual,
C. Wear business attire,
D. Wear whatever clothes they feel comfortable in.
Please focus only on any details in your data relevant to one specific option and clearly summarize its findings and rationale. Provide examples if necessary.

---

**Manager agent prompt (planner)**

You are the Manager Agent. You will receive responses compiled from several Worker Agents, each analyzing different portions of input data related to dressing for a master's graduation. Your task is to review and synthesize these individual analyses into a comprehensive overview. Consider each worker agent's insights on the various dress options:
A. Tuxedo or robe,
B. Business casual,
C. Business attire,
D. Comfortable clothes.
Highlight the most popular trends and any notable reasons supporting these choices. Pay attention to diversity in preferences, cultural considerations, and any statistical or anecdotal evidence provided by worker agents. Your final output should deliver a clear synthesis on how users most likely plan to dress for this occasion, potentially indicating preferences, influences, and recommendations.
Responses from candidates: **{response_from_workers}**

---

## E.3 EXAMPLE: MATH

---

**Raw prompt (Math)**

**{context}** (long list of numbers)
What is the **2nd smallest number**?

---

---

**Worker agent prompt (planner)**

You are a worker agent tasked with analyzing a portion of a list of numbers. From the chunk of numbers provided below, identify and return the **two smallest numbers** in ascending order.
Here is your list:
**{text_chunk}**
Please output the two smallest numbers you find in your assigned list.

---

**Manager agent prompt (planner)**

You are the manager agent tasked with synthesizing results from multiple worker agents. Each worker agent has provided the **two smallest numbers** from their respective chunks of a larger list. Your task is to compile these results and determine the **2nd smallest number** from the combined outputs of all worker agents.
Responses from worker agents:
**{response_from_workers}**
Please analyze these results and provide the **2nd smallest number** from the original list.

---

## F    MODEL NOISE RESULTS

This section provides the detailed numerical data that form the basis for Figure 2 presented in Section 5.2 of the main paper. Figure 2 illustrates the performance degradation of single-agent Large Language Models (LLMs) as the input context length increases, a phenomenon consistent with length-induced model degradation. Table 2 tabulates the specific accuracy scores for the evaluated models (gpt4o, gpt4omini, llama3b, and llama70b) on the Key-Value Retrieval (KV) and Math Find Number (Math) tasks, across a range of input lengths from 1K to 128K tokens. These raw scores directly support the visualizations and analysis concerning the growth of model error with increasing context size discussed in the main text.

Table 2: Model noise results. Scores for both tasks are accuracy.

| Task | Task length | gpt4o | gpt4omini | llama3b | llama70b |
|------|------------|-------|-----------|---------|----------|
| KV   | 1K         | 1.00  | 1.00      | 1.00    | 1.00     |
| KV   | 2K         | 1.00  | 1.00      | 0.98    | 1.00     |
| KV   | 4K         | 1.00  | 1.00      | 0.95    | 1.00     |
| KV   | 8K         | 1.00  | 1.00      | 0.88    | 1.00     |
| KV   | 16K        | 1.00  | 1.00      | 0.66    | 1.00     |
| KV   | 32K        | 1.00  | 0.99      | 0.24    | 1.00     |
| KV   | 64K        | 1.00  | 0.86      | 0.01    | 0.91     |
| KV   | 128K       | 1.00  | 0.60      | 0.01    | 0.15     |
| Math | 1K         | 0.67  | 0.71      | 0.23    | 0.63     |
| Math | 2K         | 0.68  | 0.65      | 0.19    | 0.65     |
| Math | 4K         | 0.71  | 0.64      | 0.19    | 0.66     |
| Math | 8K         | 0.69  | 0.61      | 0.10    | 0.55     |
| Math | 16K        | 0.62  | 0.54      | 0.06    | 0.55     |
| Math | 32K        | 0.63  | 0.51      | 0.06    | 0.52     |
| Math | 64K        | 0.54  | 0.37      | 0.04    | 0.39     |
| Math | 128K       | 0.33  | 0.11      | 0.00    | 0.09     |

## G    TASK NOISE RESULTS

This section contains the detailed numerical results corresponding to Figure 3 in Section 5.3 of the main paper. Figure 3 illustrates the joint effect of cross-chunk dependency (the task term, $\mathcal{L}_{\text{task}}$) and length-induced degradation (the model term, $\mathcal{L}_{\text{model}}$), showcasing how performance varies for

different LLM agents on 128K-token length tasks when the effective model context length (i.e., chunk size in the D&C setting) is altered. Table 3 lists the precise performance metrics (accuracy, F1 score, or ROUGE score, depending on the task) for the models gpt4omini, llama70b, and qwen72b. These scores are provided for all six benchmark tasks (KV, Math, QA-IB, QA-LB, Sum, Char) across various model context lengths ranging from 1K to 64K tokens (when processing a total input of 128K tokens). This data underpins the analysis of the three noise regimes and the conditions under which D&C approaches are advantageous, as discussed in the main text.

Table 3: Task noise results. Scores for all tasks are accuracy except sum which is ROUGE and qaib which is F1 score.

| Model | Context | Char | KV | Math | QA-IB | QA-LB | Sum |
|---|---|---|---|---|---|---|---|
| gpt4omini | 1000 | 0.12 | 0.99 | 0.49 | 0.22 | 0.33 | 0.08 |
| gpt4omini | 2000 | 0.12 | 0.99 | 0.50 | 0.27 | 0.44 | 0.11 |
| gpt4omini | 4000 | 0.15 | 0.99 | 0.55 | 0.39 | 0.44 | 0.15 |
| gpt4omini | 8000 | 0.17 | 0.99 | 0.54 | 0.36 | 0.48 | 0.14 |
| gpt4omini | 16000 | 0.15 | 1.00 | 0.50 | 0.37 | 0.44 | 0.15 |
| gpt4omini | 32000 | 0.18 | 1.00 | 0.55 | 0.42 | 0.46 | 0.11 |
| gpt4omini | 64000 | 0.16 | 0.98 | 0.45 | 0.39 | 0.38 | 0.11 |
| llama70b | 1000 | 0.04 | 0.99 | 0.47 | 0.44 | 0.15 | 0.16 |
| llama70b | 2000 | 0.05 | 0.96 | 0.49 | 0.55 | 0.36 | 0.19 |
| llama70b | 4000 | 0.05 | 0.99 | 0.50 | 0.55 | 0.46 | 0.23 |
| llama70b | 8000 | 0.07 | 0.98 | 0.50 | 0.56 | 0.33 | 0.28 |
| llama70b | 16000 | 0.07 | 1.00 | 0.57 | 0.63 | 0.26 | 0.24 |
| llama70b | 32000 | 0.13 | 1.00 | 0.53 | 0.54 | 0.28 | 0.23 |
| llama70b | 64000 | 0.17 | 0.91 | 0.39 | 0.41 | 0.31 | 0.21 |
| qwen72b | 1000 | 0.08 | 0.98 | 0.33 | 0.32 | 0.31 | 0.08 |
| qwen72b | 2000 | 0.07 | 0.95 | 0.46 | 0.40 | 0.28 | 0.18 |
| qwen72b | 4000 | 0.07 | 0.97 | 0.41 | 0.44 | 0.41 | 0.29 |
| qwen72b | 8000 | 0.15 | 0.98 | 0.40 | 0.48 | 0.54 | 0.23 |
| qwen72b | 16000 | 0.15 | 1.00 | 0.36 | 0.48 | 0.46 | 0.18 |
| qwen72b | 32000 | 0.13 | 0.88 | 0.31 | 0.42 | 0.46 | 0.19 |

# H    FRAMEWORK UTILITY OF PREDICTIVE PARAMETER ESTIMATION

An important measure of a theoretical framework's value lies in its empirical utility, extending beyond explanatory power to offer practical benefits. This section discusses both the explanatory strengths of our noise decomposition framework and its predictive utility in efficiently determining optimal parameters for Divide and Conquer (D&C) strategies.

## H.1    EXPLANATORY POWER

Our framework provides a principled understanding of why and when D&C methods succeed or fail in the context of long-document processing by Large Language Models (LLMs). By decomposing the overall error into distinct terms—task ($\mathcal{L}_{\text{task}}$), model ($\mathcal{L}_{\text{model}}$), and aggregator ($\mathcal{L}_{\text{agg}}$)—the framework facilitates the disentanglement of various contributing factors to performance. This structured understanding has been instrumental in deriving the key insights presented throughout this paper, such as the conditions favoring chunking and the impact of accelerated performance degradation with length.

## H.2    PREDICTIVE UTILITY FOR OPTIMAL CHUNK SIZE ESTIMATION

A practical challenge in deploying D&C methods is the selection of an optimal chunk size, which often requires costly exhaustive grid searches over various configurations. We conducted new experiments to evaluate whether our framework's insights could guide a more efficient estimation

of this parameter, particularly in regimes dominated by model noise (Regime 3 in Section 3.6), where chunk size significantly impacts performance. In other regimes (Regime 1 and Regime 2), performance tends to be less sensitive to chunk size variations, making this estimation less critical.

We focused on tasks where model noise is a dominant factor (QA-IB and Summarization, as identified in Figure 3) and attempted to estimate the optimal chunk size by evaluating performance on a very small number of randomly selected document samples (3, 5, or 10 samples) for each potential chunk size configuration. The D&C performance achieved with the chunk size selected via this minimal sampling approach was then compared to the optimal performance found through an exhaustive grid search over all samples for all chunk sizes.

The results for the QA-IB and Summarization (Sum) tasks are presented in Table 4 and Table 5, respectively. These results demonstrate that even with minimal sampling (as few as 3-5 samples per configuration), it is possible to identify chunk sizes that yield near-optimal D&C performance. This suggests a pathway to significant computational savings when setting up D&C pipelines for new tasks or models, by avoiding exhaustive searches. For example, in the QA-IB task with 'llama70b', using just 5 samples per chunk size configuration allowed us to identify a 16K chunk size yielding a score of 0.63, identical to the optimal score and chunk size found via exhaustive search. Similarly, for 'qwen72b' on the Summarization task, 5 samples were sufficient to find the optimal 4K chunk size and achieve the optimal score of 0.29.

Table 4: Predictive Utility on QA-IB: Optimal Chunk Size Estimation (Score (Optimal Chunk Size Found)). Total task length 128K tokens.

| Model | 3-sample | 5-sample | 10-sample | Optimal after Exhaustive Search |
|---|---|---|---|---|
| gpt4omini | 0.38 (64K) | 0.42 (32K) | 0.42 (32K) | 0.42 (32K) |
| llama70b | 0.55 (2K) | 0.63 (16K) | 0.63 (16K) | 0.63 (16K) |
| qwen72b | 0.40 (2K) | 0.48 (16K) | 0.48 (8K) | 0.48 (8K & 16K) |

Table 5: Predictive Utility on Summarization (Sum): Optimal Chunk Size Estimation (Score (Optimal Chunk Size Found)). Total task length 128K tokens.

| Model | 3-sample | 5-sample | 10-sample | Optimal after Exhaustive Search |
|---|---|---|---|---|
| gpt4omini | 0.15 (16K) | 0.14 (8K) | 0.14 (8K) | 0.15 (4K & 16K) |
| llama70b | 0.23 (32K) | 0.24 (16K) | 0.28 (8K) | 0.28 (8K) |
| qwen72b | 0.23 (8K) | 0.29 (4K) | 0.29 (4K) | 0.29 (4K) |

RATIONALE FOR EFFICIENT ESTIMATION

The feasibility of accurately estimating optimal chunk sizes from sparse samples can be understood through our framework's characterization of length-induced degradation. We posit that model length-induced degradation exhibits superlinear growth with context length $L = ||x||$. It is reasonable to infer that this underlying length-induced degradation function $g(L)$ is, in practice, often (or can be approximated as) monotonic with respect to $L$ over relevant ranges. Consequently, within the D&C paradigm, reducing the per-worker chunk size is expected to lead to a predictably monotonic (or near-monotonic) decrease in the dominant model term, $\mathcal{L}_{\text{model}}$.

When $\mathcal{L}_{\text{model}}$ dominates (and the task and aggregator terms, $\mathcal{L}_{\text{task}}$ and $\mathcal{L}_{\text{agg}}$, are relatively small or controlled), the overall D&C system error as a function of chunk size is likely to exhibit a discernible optimal region (e.g., a convex-like shape or a region where decreasing length-induced degradation is balanced by slowly increasing decomposition/aggregation overhead) rather than fluctuating randomly and unpredictably. Therefore, a small number of samples across different chunk sizes can provide sufficient information to sketch the rough contour of this error curve, allowing for the identification of the approximate location of this optimal trade-off point without exhaustive evaluation.

## I    THE IMPACT OF OVERLAP IN CHUNKING STRATEGIES

In our main analysis, we primarily considered non-overlapping chunks. To further explore variations in chunking methodologies, we investigated the impact of introducing a modest overlap between adjacent chunks. This strategy is sometimes employed with the aim of mitigating potential information loss at chunk boundaries, which could otherwise increase cross-chunk dependency effects (the task term, $\mathcal{L}_{\text{task}}$). We conducted experiments using the Llama-70B model on several 128K-token long-context tasks, comparing non-overlapping chunks with chunks having a 1K token overlap (base chunk sizes were 4K and 16K tokens before overlap).

The results are presented in Table 6. The inclusion of a 1K token overlap led to mixed and generally

Table 6: Impact of 1K Token Overlap on Llama-70B Performance (128K Tasks)

| Llama-70B Strategy | KV | QA-IB | Sum | Char |
|---|---|---|---|---|
| 16K chunks, no overlap | 1.00 | 0.63 | 0.24 | 0.07 |
| 16K chunks, 1K overlap | 1.00 | 0.54 | 0.25 | 0.07 |
| 4K chunks, no overlap | 0.99 | 0.55 | 0.23 | 0.05 |
| 4K chunks, 1K overlap | 1.00 | 0.53 | 0.19 | 0.05 |

marginal changes in performance. While slight improvements were noted in some specific instances, other cases showed minor degradations or no discernible effect. This suggests that while a small overlap might offer a limited benefit for certain local cross-chunk dependencies, it does not consistently or substantially alter the overall performance trade-offs identified by our framework for these tasks. Moreover, extensive overlap could introduce processing redundancies or even conflicting information for the aggregator, potentially increasing aggregation errors (the aggregator term, $\mathcal{L}_{\text{agg}}$). For the configurations tested, the benefits of this overlap strategy did not appear to consistently outweigh the increased complexity or computational cost.

## J    COMPARATIVE ANALYSIS WITH RETRIEVAL-AUGMENTED GENERATION (RAG)

Retrieval-Augmented Generation (RAG) represents a prominent alternative for addressing long-context tasks by first retrieving presumptively relevant segments of the input, which are then processed by the LLM. To situate our Divide and Conquer (D&C) framework in relation to this approach, we performed a comparative analysis. We applied RAG to the single-shot baseline models for the 128K-token versions of our benchmark tasks, employing both BM25 (sparse retrieval) and 'all-mpnet-base-v2' embeddings (dense retrieval). Based on the task query, the RAG pipeline retrieved document sections that were then provided as context to the LLM.

The performance of RAG is detailed in Table 7. For several tasks, RAG was notably less effective than

Table 7: Comparison of D&C (Implicit via "noRAG" baseline) and RAG Strategies

| Model | Method | KV | QA-IB | QA-LB | Sum | Char |
|---|---|---|---|---|---|---|
| gpt4omini | noRAG (Baseline) | 0.60 | 0.23 | 0.31 | 0.13 | 0.19 |
| | RAG (BM25) | 0.79 | 0.13 | 0.23 | 0.12 | 0.11 |
| | RAG (mpnet) | 0.73 | 0.19 | 0.27 | 0.13 | 0.13 |
| llama70b | noRAG (Baseline) | 0.15 | 0.56 | 0.23 | 0.19 | 0.18 |
| | RAG (BM25) | 0.81 | 0.14 | 0.23 | 0.15 | 0.10 |
| | RAG (mpnet) | 0.77 | 0.38 | 0.26 | 0.15 | 0.14 |

both our D&C methodology and the original single-shot baseline. Tasks such as summarization or character inference, which often require a global understanding or synthesis of diffuse information not easily targeted by a simple query, proved challenging for the retrieval step. Inaccurate or incomplete retrieval frequently provided the LLM with a partial or skewed view of the overall context, leading to

degraded performance. These findings highlight the sensitivity of RAG to retrieval quality and its applicability to tasks where key information is sparse or not easily queryable. Our D&C approach, by processing the entirety of the text through structured decomposition, can offer greater robustness for such scenarios.

## K    RELATION TO ARCHITECTURAL CONTEXT EXTENSION METHODS

Alongside algorithmic approaches like D&C, significant research efforts focus on extending the native context window capabilities of LLM architectures, often through modifications to positional encodings (e.g., RoPE scaling) or continued training on longer sequences. It is noteworthy that one of our primary baseline models, QWen2.5-72B-Instruct, already incorporates such advanced context extension techniques and has undergone training with them.

To further understand the landscape, particularly regarding training-free extension methods, we examined the behavior of Llama-2 (a model with a nominal 4K token context window) when its context was extended to 32K tokens via RoPE scaling, without additional fine-tuning. The results are shown in Table 8. The data indicate a decline in performance as the input length significantly

Table 8: Performance of Llama-2 with RoPE Scaled Context Extension (Training-Free)

| Model | Tested CTX | KV | QA-IB | Char |
|---|---|---|---|---|
| Llama2-4K | 4K | 0.47 | 0.23 | 0.00 |
| Llama2-32K (Scaled) | 8K | 0.42 | 0.19 | 0.00 |
| | 16K | 0.23 | 0.17 | 0.03 |
| | 32K | 0.12 | 0.13 | 0.00 |

surpassed the model's original training parameters. This aligns with broader observations in the field (e.g., the LongRoPE study) suggesting that while training-free scaling offers some benefit, achieving robust performance at substantially extended context lengths typically requires further training or architectural adaptation. The D&C framework presented in this paper is complementary, providing a strategy to handle extensive sequences that can be applied irrespective of a model's inherent maximum context length.

## L    CONNECTION TO EFFECTIVE CONTEXT LENGTH RULER BENCHMARKS

The efficacy of any long-context strategy is intrinsically linked to the underlying LLM's capabilities. Benchmarks like Ruler aim to quantify the "effective context length" of models, often focusing on retrieval-based tasks. To assess the generalizability of our noise decomposition framework across a spectrum of models and task types beyond simple retrieval, we conducted additional evaluations. We selected three models with differing architectural characteristics and reported long-context capabilities: Mistral-7B-Instruct-v0.2, Qwen1.5-72B-Chat, and GPT-4-Turbo.

These models were evaluated on several of our benchmark tasks with varying input context lengths. The performance patterns are summarized in Table 9. While a longer effective context is generally advantageous, the specific manifestation of the model term ($\mathcal{L}_{\text{model}}$) and the task term ($\mathcal{L}_{\text{task}}$) remains highly task-dependent. Complex reasoning or generation tasks can stress long-context capabilities differently than retrieval tasks. These observations reinforce the utility of our noise decomposition framework as an analytical tool. The framework helps explain performance variations by considering the interplay of the task's intrinsic decomposability, the model's confusion threshold with increasing context, and the aggregator's ability to synthesize information, providing insights that extend beyond a single metric of effective context length.

## M    FRAMEWORK IMPLEMENTATION IN MORE DETAILS

We present a simple implementation of the framework as in Figure 1. Our implementation consists of three parts: a planner for automated query planning, numerous worker agents, and single manager

Table 9: Performance of Diverse Models with Varying Context Lengths

| Model | Ruler CTX (Approx.) | Tested CTX | KV | QA-LB | Sum |
|---|---|---|---|---|---|
| Mistral-7B | 16K | 4K | 0.03 | 0.230 | 0.090 |
| | | 8K | 0.00 | 0.290 | 0.090 |
| | | 16K | 0.11 | 0.170 | 0.090 |
| Qwen1.5-72B-Chat | 32K | 4K | 0.92 | 0.360 | 0.110 |
| | | 8K | 0.97 | 0.260 | 0.130 |
| | | 16K | 0.95 | 0.310 | 0.110 |
| | | 32K | 0.93 | 0.270 | 0.150 |
| GPT-4 (Turbo) | 64K+ | 4K | 1.00 | 0.431 | 0.088 |
| | | 8K | 1.00 | 0.461 | 0.091 |
| | | 16K | 1.00 | 0.435 | 0.092 |
| | | 32K | 0.98 | 0.399 | 0.085 |
| | | 64K | 0.87 | 0.384 | 0.081 |
| | | 128K | 0.77 | 0.373 | 0.074 |

agent. For a long context task, we divide it into chunks of approximately equal length and feed each chunk to one worker agent. Thus, we need to design how worker agents process their chunks and how manager agent could synthesize the appropriate answer from the interaction.

## M.1 WORKER AGENT

In our framework, the long context sequence is divided into multiple segments, with each worker agent independently responsible for processing one specific segment. This design isolates the processing of individual segments, deliberately omitting any consideration of dependencies across segments. The goal is to simplify each worker's task by restricting its scope to a single context segment. While our implementation employs homogeneous worker agents for simplicity, this setup can be extended to accommodate heterogeneous agents with varying capacities in more complex scenarios.

## M.2 MANAGER AGENT

The manager agent is tasked with aggregating the outputs generated by the worker agents. It analyzes the responses received and synthesizes a unified, refined output. For simplicity in our current design, we use the same model architecture for the manager agent as for the worker agents. However, this design can be adapted to allow for specialized managerial processing in more advanced applications.

## M.3 PLANNER

The planner plays a central role in the coordination of the system. While we have outlined the basic structure of worker and manager agents, the specific responsibilities of each agent will vary depending on the task and the data. Instead of manually assigning tasks to the agents, we introduce a planner that autonomously determines the optimal distribution of jobs between the worker and manager agents. The planner is given the original task description (e.g., "identify the second largest number in this list") and is responsible for generating appropriate prompts for each agent, taking into account the segmentation of the long sequence. Moreover, human-designed agent workflows typically involve a trial-and-error approach: initial job prompts are tested on a small dataset, the failures are analyzed, and the design is iteratively refined. We integrate this iterative process into the planner by having it perform an initial zero-shot job assignment, followed by an evaluation on holdout validation data. The planner receives feedback on mispredicted cases and revises the job prompts accordingly. While this iterative process can be repeated to improve performance, there is a risk of overfitting if carried out for too long. In our experiments, we iterate only once to avoid trivial problems.

# N   COMPUTATIONAL COST AND LATENCY OF D&C VS. SINGLE-PASS

**Setup and notation.**   Let an input have length $T$ tokens and be split into $n$ equal chunks. Denote

- $T_{\text{single}}(T)$: latency for a single, large model to process the full input,
- $T_{\text{dc}}(T/n)$: latency for a smaller worker model to process one chunk,
- $T_{\text{manager}}(L_{\text{agg}})$: latency for the manager/aggregator to process the short aggregation input of length $L_{\text{agg}}$ (concatenated worker outputs/prompts), where typically $L_{\text{agg}} \ll T$.

We assume worker calls are issued in parallel up to the available concurrency of the deployment.

## N.1   LATENCY

**Single-pass wall-clock latency.**

$$\text{Latency}_{\text{single}} \; = \; T_{\text{single}}(T). \tag{7}$$

**D&C wall-clock latency with parallel workers.**

$$\text{Latency}_{\text{D\&C}} \; \approx \; T_{\text{dc}}(T/n) \; + \; T_{\text{manager}}(L_{\text{agg}}). \tag{8}$$

**When is D&C faster?**

$$\boxed{T_{\text{single}}(T) \; > \; T_{\text{dc}}(T/n) \; + \; T_{\text{manager}}(L_{\text{agg}})} \tag{9}$$

This inequality is often satisfied in practice because (i) per-call latency grows with input length, so shrinking from $T$ to $T/n$ reduces each worker's processing time, and (ii) $L_{\text{agg}}$ is short, so the manager adds a small, near-constant overhead. Crucially, D&C does *not* require $n$ sequential LLM calls; with parallelization the critical path is one worker pass plus a single manager pass.

**Practical caveats.**   The achievable speedup depends on available parallel capacity (API/cluster concurrency), network overhead, and the manager prompt design. Excessive chunk overlap or verbose worker outputs increase $L_{\text{agg}}$ and can erode gains.

## N.2   MONETARY COST

Let $p_{\text{in/out}}^{\text{big}}$ be per-token prices for the single, flagship model; $p_{\text{in/out}}^{\text{small}}$ for the smaller worker model; and $p_{\text{in/out}}^{\text{mgr}}$ for the manager. Let $|y|$ be the final output length, $|y_i|$ each worker's output length, and $L_{\text{agg}}$ the manager's input length.

**Single-pass cost.**

$$\text{Cost}_{\text{single}} \; \approx \; p_{\text{in}}^{\text{big}} \, T \; + \; p_{\text{out}}^{\text{big}} \, |y|. \tag{10}$$

**D&C cost.**

$$\text{Cost}_{\text{D\&C}} \; \approx \; p_{\text{in}}^{\text{small}} \, T \; + \; p_{\text{out}}^{\text{small}} \sum_{i=1}^{n} |y_i| \; + \; p_{\text{in}}^{\text{mgr}} \, L_{\text{agg}} \; + \; p_{\text{out}}^{\text{mgr}} \, |y|. \tag{11}$$

**Token accounting and implication.** With non-overlapping chunks, the dominant input mass remains $T$ in both pipelines. D&C adds only a small extra budget for $L_{\text{agg}}$ and $\sum_i |y_i|$ (kept short by structured worker outputs). Therefore, when $p_{\text{in/out}}^{\text{small}} \ll p_{\text{in/out}}^{\text{big}}$ (using compact/open models as workers and a lightweight manager), D&C is typically cheaper while processing roughly the same number of dominant input tokens.

**Practical caveats.**   Chunk overlap, verbose worker outputs, retries, or using a large manager can increase $L_{\text{agg}}$ and $\sum_i |y_i|$. These are controlled by (i) minimizing overlap, (ii) constraining worker output schemas, and (iii) keeping the manager focused on short structured inputs.

### N.3 SUMMARY OF THE TRADE-OFF

- **Latency:** With parallel workers, D&C reduces the critical path from processing $T$ once to processing a single chunk of size $T/n$ plus a short aggregation step (Eq. 9).

- **Cost:** Because the dominant token mass $T$ is similar in both methods, shifting those tokens to cheaper worker models and paying only a small aggregation overhead typically lowers total cost.

- **Key condition:** Short, structured worker outputs (small $L_{\text{agg}}$) and sufficient parallel capacity make $T_{\text{single}}(T) > T_{\text{dc}}(T/n) + T_{\text{manager}}(L_{\text{agg}})$ and $\text{Cost}_{\text{D\&C}} < \text{Cost}_{\text{single}}$ realistic for long inputs.

These considerations complement the performance-centric analysis in the main paper: when model noise grows with context length and cross-chunk synergy is moderate, D&C can simultaneously improve quality, reduce wall-clock time, and lower cost relative to a single-pass large-model run.

## O WHY STATIC EQUAL-LENGTH CHUNKING?

For clear analysis and diagnosis, we adopt a *control-variable* design: we set the *chunk size* equal to the *per-chunk context length* and use equal-length, non-overlapping splits. This way, the performance change as chunk size varies can be primarily attributed to the *length effect* (model noise), rather than multiple moving parts at once.

**Why not use adaptive segmentation in the main experiments?** Adaptive policies (variable lengths, semantic boundaries, routing) *simultaneously* change:

- the **length distribution** across chunks (affecting how model noise scales with length),
- the **boundary placement/routing** relative to dependencies (affecting task noise),
- the **structure of worker outputs** (affecting aggregator noise).

When several factors vary together, the diagnostic curve (*performance vs. chunk size*) becomes hard to interpret, and we can no longer cleanly read off the relationship between per-chunk length and model confusion, nor the relative contribution of the three noise sources.

**Benefits of equal-length splitting (control of variables)** Equal-length, non-overlapping splits concentrate the degree of freedom onto the per-chunk length $\ell$, which:

- enables a direct estimation/visualization of how model noise depends on $\ell$ (the basis for our read of Fig. 2/3),
- stabilizes comparisons between D&C and single-shot across different $\ell$,
- makes the diagnosis of "model-noise dominated" vs. "task-noise dominated" regimes more reliable.

Empirically, we also observed that small overlaps only yield *marginal* gains in our setting, suggesting the conclusions are not artifacts of brittle boundaries.

**The framework does not preclude adaptivity** Our decomposition naturally accommodates adaptive segmentation; when diagnostics indicate high task noise (strong cross-chunk dependencies), adaptive strategies can be useful in engineering practice. However, the focus of this paper is first to *identify* the length-driven effect under controlled conditions. In other words, we do not reject adaptivity; we fix chunk lengths to keep variables controlled so that the analysis remains interpretable and reproducible.

