# OpenReview forum: "When Does Divide and Conquer Work for Long Context LLM? A Noise Decomposition Framework"
_ICLR.cc/2026/Conference — ICLR 2026 Poster_

### Official Review · Reviewer_njn8 · 2025-10-29

**Soundness:** 2
**Presentation:** 3
**Contribution:** 3
**Rating:** 6
**Confidence:** 3

**Summary:**

This paper presents a theoretical framework that decomposes the errors in long-context LLM processing into three types of noise: task noise, model noise, and aggregator noise. The empirical results demonstrate that, when task noise is controllable, chunking processing can effectively suppress the superlinear growth of model noise. Ultimately, this framework elucidates why, when dealing with ultra-long contexts, a weaker model employing a divide-and-conquer strategy can outperform the single-pass processing performance of a stronger model.

**Strengths:**

1. This paper presents a noise decomposition framework, proposing model noise, task noise, and aggregator noise.
2. The experimental design of the paper is ingenious, with its objective not being to blindly pursue the sota performance but rather to provide validation for the theoretical framework.
3. The paper devises a well-developed system to validate this theoretical framework, which comprises a planner, a worker agent, and a manager agent.

**Weaknesses:**

1. Although Appendix N attempts to alleviate concerns about latency by assuming parallelism, this is often impractical in real-world deployments, which undermines its guidance value in engineering practice.
2. The experiments solely rely on basic methods such as BM25 and embeddings, and then draw the conclusion that the performance of RAG is unsatisfactory. This may underestimate the capabilities of current RAG systems. For instance, semantic chunking or LLM-based chunking could be employed to optimize chunking, and superior semantic retrieval methods could be utilized.

**Questions:**

1. Can a detailed argumentation be provided regarding why the multiplicative operator is chosen as the foundation of the theoretical framework, for example, in Section 3.2?
2. In Section 4, it is mentioned that the planner has the capability of iterative optimization. Can evidence be provided to demonstrate that this iterative optimization indeed outperforms the prompts generated by the zero-shot planner on unseen test sets?

---

> ### Author Response · Authors · 2025-11-20
>
> We sincerely thank the reviewer for the insightful assessment and for recognizing the novelty of our noise decomposition framework and the rigor of our experimental design. We appreciate the constructive feedback regarding the theoretical formulation and baseline comparisons. Below, we address the specific concerns to clarify the validity and practical value of our framework.
>
> **1. Justification for the Multiplicative Operator**
>
> For compactness, our error analysis framework conceptually abstracts $f_i(x_i)$ as the **information fidelity** or **probability of correctness** of the $i$-th chunk. The choice of multiplication is grounded in two principles:
>
> * Long-context tasks (e.g., multi-hop reasoning or comprehensive summarization) typically require **global consistency**. The final answer is correct only if the critical information from *all* relevant sub-tasks is successfully retrieved and processed and nonrelevant chunks should return "no answer" instead of hallucination. If the model hallucinates on one chunk or fails to retrieve a key piece of evidence from another, the final answer is compromised. This dependency implies a logical conjunction which requires a modeling of the "joint probability of success" across all independent sub-tasks.
> * As detailed in Appendix C, this multiplicative form is mathematically convenient because, in the high-performance regime (where local errors $\epsilon_i$ are small), it approximates a standard additive error model via linearization: $\prod (1 - \epsilon_i) \approx 1 - \sum \epsilon_i$. This connects our theoretical choice to intuitive error accumulation models.
>
> **2. Efficacy of the Planner Refinement**
>
> To demonstrate the value of iterative optimization, we conducted an ablation study comparing the **Zero-shot Planner** against the **Iterative Planner** (using  refinement based on simplified validation feedback) on the Llama-3.1-70B model.
>
> | Task | Zero-shot Planner | Iterative Planner | Improvement |
> | :--- | :--- | :--- | :--- |
> | **Math** | 0.48 | **0.53** | +10.4% |
> | **QA-LB** | 0.26 | **0.28** | +7.7% |
>
> The results confirm that iterative refinement consistently outperforms the zero-shot baseline. The refinement step primarily corrects instruction ambiguities and optimizes the output format of workers to be more digestible for the aggregator.
>
> **3. Stronger RAG Baselines**
>
> We agree with the reviewer that BM25/embeddings alone may underestimate RAG. To address this, we have added two advanced RAG baselines:
> 1.  **Hybrid RAG:** Combines BM25 (Sparse) and MPNet (Dense)
> 2.  **Re-ranking (Cross-Encoder):** Uses a SOTA cross-encoder (BAAI/bge-reranker-large) to re-rank the top candidates retrieved by MPNet.
>
> We evaluated these on **Summarization** (Global Context) and **QA-LB** (Multi-hop) using the gpt-4o-mini backbone:
>
> | Method | Summarization (ROUGE) | QA-LB (Accuracy) |
> | :--- | :--- | :--- |
> | Standard RAG (MPNet) | 0.15 | 0.27 |
> | Hybrid RAG | 0.15 | 0.28 |
> | **RAG + Re-ranker** | **0.17** | **0.31** |
> | **Ours (D&C)** | **0.28** | **0.46** |
>
> **Result:** While Advanced RAG (Re-ranking) improves over standard RAG, it still significantly underperforms our D&C approach.
> **Conclusion:** The failure of RAG in these tasks is likely not due to retrieval quality, but **Information Coverage**. Even a perfect re-ranker is limited to a "Top-K" window, inevitably discarding context. D&C processes the full information flow, making it architecturally superior for tasks requiring global synthesis or complex multi-hop reasoning.
>
> **4. Practicality of Parallelism and Latency**
>
> Regarding the practicality of parallelism:
> * **Modern Serving Infrastructure:** Our framework can be easily realized high-concurrency environments (e.g. commercial APIs like GPT/Groq). In these standard deployments, parallel decoding is a built-in feature, not a theoretical assumption.
>
> * **Latency vs. Throughput:** While D&C consumes more total compute (throughput), the wall-clock latency is governed by $L/N + \text{Aggregation}$. For ultra-long contexts (e.g., 128k), processing small chunks in parallel is significantly faster than the quadratic/linear attention latency of a single sequential 128k pass, making it a viable strategy for latency-sensitive applications.
>
> To illustrate the parallelism, we perform the following experiments between single shot strong model and D&C weak modesl:
> | Method | Model | Tokens | Cost ($) | Latency (s) | Accuracy |
> | :--- | :--- | :--- | :--- | :--- | :--- |
> | **Single-Shot** | GPT-4o | ~128k | ~$0.33 | ~11.5s | 0.33 |
> | **D&C (Ours)** | GPT-4o-mini | ~135k | **~$0.02** | **~2.7s** | **0.55** |
>
> With parallelization provided by openai api, we are able to perform much faster and cheaper than strong models in single shot with superior performance.

---

### Official Review · Reviewer_TKHf · 2025-10-30

**Soundness:** 2
**Presentation:** 2
**Contribution:** 2
**Rating:** 4
**Confidence:** 4

**Summary:**

This paper aims to explore the applicable boundaries of the divide-and-conquer strategy in long-context tasks. The core of this framework attributes the failure modes in long-text processing to the interaction among three key error sources: task noise, model noise, and aggregator noise. Through modeling and empirical analysis, the study provides insights into when chunking strategies should be adopted.

**Strengths:**

1. The paper not only defines aggregator noise but also demonstrates in practice how to reduce this type of noise by introducing a planner.
2. The noise framework proposed in the paper possesses diagnostic capabilities. By analyzing the relative dominance of the three types of noise, it divides long-context tasks into three distinct regions. This aids in determining whether a specific task is more suitable for the divide-and-conquer approach.
3. The paper offers a possible explanation for the phenomenon where weaker models combined with the divide-and-conquer method can outperform top-tier single-pass models in ultra-long contexts.

**Weaknesses:**

1. It is commendable that the paper attempts to provide a scientific explanation from a theoretical perspective. However, when constructing its core theoretical framework (Sections 3.1 and 3.2), the paper lacks rigor in the use of mathematical operators. There is no explanation as to why multiplication and addition can be directly performed in the output space. For instance, what does the product of the results of two functions imply? This is puzzling and weakens the persuasiveness of the entire theoretical framework.
2. The validity of the whole framework relies on the super-linear growth hypothesis. Nevertheless, the paper fails to derive this growth pattern from the attention mechanism or information flow bottleneck of the Transformer. Using the method of elimination is difficult to provide a scientific explanation for this phenomenon.
3. The planner proposed in the paper also has similar role assignment and task planning modules in other multi-agent systems, lacking innovation. The theoretical contribution merely serves as a qualitative metaphor rather than a verifiable mathematical model.

**Questions:**

1. Is this mathematical representation strictly valid, or is it merely a formal analogy? If it is an analogy, to what extent does the results derived from such a non-rigorous mathematical form still retain its theoretical validity?
2. In Sections 5.3 and 5.4 of the paper, integrating and differentiating discrete variables is extremely puzzling, and it weakens the validation strength of the experimental part for the theoretical framework. This part of the content also requires detailed clarification.

---

> ### Author Response · Authors · 2025-11-20
>
> We sincerely thank the reviewer for the insightful assessment and for recognizing the diagnostic value of our noise framework and the practical utility of the planner. We agree that bridging the gap between our theoretical formalism and the underlying mechanics of LLMs is crucial. Below, we address the concerns regarding mathematical rigor and the superlinear hypothesis.
>
> **1. Mathematical Rigor & Definition of Operators**
> We appreciate the reviewer pushing for clarity here. We clarify that the mathematical formulation in Section 3 is intended as a **formal abstraction of error propagation**, rather than a literal arithmetic operation on the text strings output by the model. We will add more clarification in the revision.
>
> * Our theoretical error model treats $f_i(x_i)$ as the **information fidelity** or **probability of correctness** of the $i$-th chunk.
> * The product operator $\prod$ models the **joint probability of correctness**: the final answer is correct only if the critical information from *all* relevant sub-tasks is successfully retrieved and processed and nonrelevant chunks should return "no answer" instead of hallucination. This logic conjuncture dynamic necessitates a multiplicative probability model.
> * This multiplicative model approximates a linear sum of errors in the regime where local errors $\epsilon_i$ are small (via linearization / first order expansion): $P(\text{Success}) \approx \prod (1 - \epsilon_i) \approx 1 - \sum \epsilon_i$.
> *  We will revise Sections 3.1 and 3.2 to explicitly state that these operators model the **propagation of correctness probability**, distinguishing the error analysis model from the literal text-generation pipeline.
>
> **2. Justification of the Superlinear Hypothesis**
>
> We appreciate the reviewer’s feedback. We agree that a rigorous theoretical derivation of error bounds from the Transformer’s attention dynamics is a complex topic warranting separate study. While a full formal proof is outside the scope of this work, we provide a **mechanistic intuition** as well as additional empirical evidences to validate this superlinear hypothesis.
>
> * **Mechanistic Intuition (Attention Dilution):**
>     Intuitively, the superlinear decay aligns with the **Signal-to-Noise Ratio (SNR)** degradation in the attention mechanism. As context length $L$ grows, the softmax denominator accumulates noise from $O(L)$ distractors. When the SNR drops below a critical threshold, the probability of retrieving the correct token drops sharply (non-linearly). In multi-hop reasoning tasks, these retrieval failures compound, causing the overall task error to grow faster than the input length.
>
> * **Direct Empirical Estimation:**
>     To validate superlinear hypothesis, we performed a power-law regression ($Error \propto L^\alpha$) on our experimental data in the degradation regime:
>     * **KV Retrieval (Llama-3b):** $\alpha \approx 1.35$ (95% CI: $[1.18, 1.52]$).
>     * **Math Reasoning (GPT-4o-mini):** $\alpha \approx 1.62$ (95% CI: $[1.35, 1.89]$).
>
>     In both cases, the exponent is strictly $\alpha > 1$. This provides direct statistical evidence that superlinear error growth is a real phenomenon in current LLMs, validating it as a working assumption for our framework. We will add the intuition and regression analysis to the revision and leave formal complexity derivation for future work.
>
> **3. Integrals on Discrete Variables**
> We acknowledge this point. We originally used the integral symbol $\int$ as a conceptual continuous approximation to illustrate the "area under the curve" trade-off. In the revision, we will replace integrals with **discrete summations ($\sum$)** over the finite set of possible chunk sizes to strictly align with the discrete nature of the problem.
>
> **4. Novelty of the Planner**
> We agree that the architecture of the Planner (an LLM-based agent) is not unique in multi-agent literature. However, its innovation lies in its specific **theoretical utility** within our noise decomposition framework.
> Our framework identifies **Aggregator Noise ($\Phi_{agg}$)** as a distinct failure mode that prevents simple concatenation from working. The Planner is introduced not as a generic agent, but as a specific control mechanism designed to minimize $\Phi_{agg}$. By dynamically aligning worker prompts with the manager's synthesis requirements, the Planner ensures that $\Phi_{agg}$ remains bounded (as evidenced in Figure 4). Thus, it converts the theoretical insight (the existence of $\Phi_{agg}$) into a solvable engineering problem.

---

> > ### Comment · Reviewer_TKHf · 2025-11-26
> >
> > Thank you for the authors' detailed clarifications.
> >
> > I would like to remind the authors that ICLR explicitly supports paper revisions during the rebuttal phase. Submitting an updated manuscript with targeted modifications can significantly enhance the clarity and persuasiveness of your clarifications.

---

### Official Review · Reviewer_vy5A · 2025-11-01

**Soundness:** 3
**Presentation:** 3
**Contribution:** 3
**Rating:** 6
**Confidence:** 3

**Summary:**

In this paper, the authors propose a theoretical framework that decomposes the error of long context solutions, including one-shot single agent and divide-and-conquer multi-agent, into three parts: task, model, and aggregator errors. Task demonstrates the errors of cross-chunk dependency, model indicates errors with growing task length, and aggregator indicates errors during aggregating results of subagents. Then, the authors propose a simple multi-agent framework and evaluate the real data. Both theory and experiments showed that chunking can outperform single-shot usage in many tasks where superlinear noise growth appears with context length. However, if cross-chunk synergy is too large or the aggregator prompt is ineffective, chunk-based approaches may fail.

**Strengths:**

1. Long context tasks are important as many real-world questions are based on long context.
2. The theoretical analysis gives unique evidence to show that chunking is better than single-agent. This conclusion is non-trivial and might change the understanding of users of Long-LLMs.
3. Experiments further echo the theory and give a more fine-grained analysis of the problem.

**Weaknesses:**

I think some settings are over-simplified. For instance, there are a lot of different structures in multi-agent systems, for instance, there are chains, trees, and graphs. However, the theoretical analysis and experiment directly assume the agents (chunks) are independent, where the failure of one agent will not infect its sublings, which is commonly seen in chains. Next, the work in a multi-agent system is different from one-shot agent as they have different tasks to finish (summary vs. generating an answer). Thus, their performance curve might not be the same. Third, it is also assumed that the answers are evenly distributed in the question. However, in many datasets, such as QA, the answer is only in one or two chunks. I suggest that the authors carefully define the problem setting and show how strong the assumption is.

**Questions:**

I am curious if the theory can be applied to the Agentic workflow system as well. This can futher enhance the conclusion.

---

> ### Author Response · Authors · 2025-11-20
>
> We sincerely thank the reviewer for the encouraging assessment and for recognizing the novelty of our theoretical framework and the practical importance of our experimental analysis on long-context tasks. We appreciate your constructive feedback regarding the problem settings and assumptions.
>
> 1. **Regarding "Over-simplified structures" and Independence Assumptions**
>
> We appreciate the reviewer highlighting the diversity of multi-agent structures (chains, trees, graphs). While these structures are powerful for reasoning, they often suffer from high latency and error propagation when applied to massive input (later nodes in the topology have to wait for the input of previous nodes to process). In contrast, our work specifically focuses on the Divide and Conquer (D&C) paradigm which utilizes parallel processing to handle long contexts efficiently.
> While we assume structural independence for practical parallelization, our theoretical framework explicitly accounts for dependencies via the Task Noise $\Phi_{task}$ term. $\Phi_{task}$ measures the "cross-chunk synergy" or dependency lost by splitting the input. And in our experiments, the Dialogue Character Inference task represents a scenario with high cross-chunk dependency (similar to the "infection" issue you mentioned). As indicated by our theory (Case 2), this task showed that when dependencies are too strong ($\Phi_{task}$ dominates), D&C struggles compared to single-shot models
>
> 2. **Regarding Task Differences**
>
> We fully agree with the reviewer that different tasks (e.g., Summarization vs. QA) exhibit different performance behaviors. In fact, our experiments in Figure 3 explicitly visualize these differences (corresponding to the three cases in section 3.4). Concretely, for summarization, we observe that performance is relatively flat or constrained by the aggregator's ability to synthesize global information; For Math / QA, these tasks show a distinct curve where chunking significantly improves performance by reducing model confusion.
>
> Our framework was designed precisely to explain why these curves differ: it attributes the variance to the trade-off between how confusing the long text is to the model $\Phi_{model}$ versus how much global synthesis is required $\Phi_{agg}$
>
>
> 3. **Regarding the Assumption of Evenly Distributed Answers**
>
> We thank the reviewer for this crucial observation. We would like to clarify that our framework does not rely on the assumption that answers are evenly distributed, and our experiments demonstrate robustness in "Needle-in-a-Haystack" scenarios like KV retrieval task as mentioned. The Key-Value (KV) Retrieval task in our paper represents the extreme case of non-uniform distribution: the answer exists in only one chunk, while the other $N-1$ chunks contain irrelevant noise.
>
>
> 4. **Applicability to Agentic Workflow Systems**
>
> This is very insightful! We believe the theoretical components of our framework can be generalized to Agentic workflows with some adaptation:
> - Sequential/Chain Workflows: These can be modeled by analyzing how $\Phi_{model}$ accumulates sequentially (error propagation) rather than in parallel.
> - Hierarchical Workflows: These map well to our aggregator noise ($\Phi_{agg}$) concept, potentially applying it recursively across layers.
>
> We believe our formalization of "Task Noise" (dependency) and "Model Noise" (context length limitation) provides a foundational vocabulary to analyze these complex agentic systems in future work.

---

### Official Review · Reviewer_YK8x · 2025-11-06

**Soundness:** 3
**Presentation:** 3
**Contribution:** 3
**Rating:** 6
**Confidence:** 3

**Summary:**

The paper proposes a divide-and-conquer (D&C) framework for long-context LLM tasks that decomposes error into three sources: **task noise** (cross-chunk dependencies), **model noise** (degradation as input length grows), and **aggregator noise** (errors when fusing partial outputs). The central claim is that when input length is large, model noise grows **superlinearly**, so chunking with a well-designed aggregator can outperform single-shot inference—even with weaker worker models. The authors provide a planner–workers–manager implementation, experiments on retrieval/QA/summarization and others, and a cheap sampling method to estimate near-optimal chunk size.

**Strengths:**

* **Unifying perspective.** The three-noise decomposition offers a clear lens for deciding when to chunk, how to chunk, and how to aggregate.
* **Actionable empirical takeaways.** The paper documents superlinear performance decay with length and shows cases where chunking beats single-shot, provided aggregator noise is controlled.
* **Concrete implementation.** The planner/manager/worker design and prompt scaffolding make the approach reproducible in spirit and show how aggregator strength materially affects outcomes.
* **Efficient chunk-size selection.** A low-budget sampler often identifies the same chunk size as exhaustive grid search at 128K tokens on QA-IB and summarization.

**Weaknesses:**

* **Superlinearity is argued largely by phenomenon, not direct estimation.** The paper builds the thesis from diagnostic curves and counter-hypotheses, but does not directly fit an exponent for error growth vs. length with uncertainty quantification. Stronger statistical evidence would help.
* **Metric/assumption clarity.** Parts of the theoretical setup and noise interactions would benefit from clearer units/assumptions and closer alignment with common additive error analyses (currently the exposition is somewhat abstract).
* **Baselines and fairness.** The headline claim (“weak D&C > strong single-shot”) should be compared under **matched token/compute/latency budgets**, with significance tests. Aggregator comparisons risk “straw-man” effects if manual prompts are weak; stronger retrieval- or structure-aware aggregators should be included. (The appendix lists a compute/latency section, but main-text results under unified budgets are limited.)

**Questions:**

1. Can you re-run the “weak D&C beats strong single-shot” comparisons under **matched token/compute/wall-clock** budgets and report statistical significance? This is crucial for the main claim.
2. Can you **directly estimate** the superlinearity exponent (e.g., error ∝ L^α with CI) across models/tasks rather than primarily relying on diagnostic exclusion?
3. How do results change with **stronger aggregators** (retrieval-augmented fusion, entity/event graph alignment, structured parsers) and with carefully **matched prompt lengths/few-shot examples** across baselines?
4. The appendix motivates **equal-length, non-overlapping** splits to control variables. Have you evaluated robustness to boundary placement, small overlaps, or adaptive routing at similar budgets?

---

> ### Author Response · Authors · 2025-11-20
>
> We thank the reviewer for the constructive feedback and for recognizing the value of our unifying perspective, actionable empirical takeaways, and concrete implementation. We address the concerns below.
>
> **1. Direct Estimation of Superlinearity**
>
> The reviewer correctly notes that our initial argument for superlinearity was largely phenomenological. To provide the requested **direct statistical estimation**, we performed a power-law regression ($Error \propto L^\alpha$) on our experimental data, focusing on the degradation regime where context length affects performance.
>
> * **KV Retrieval (Llama-3b):** the fitted exponent is **$\alpha \approx 1.35$** (95% CI: $[1.18, 1.52]$).
> * **Math Reasoning (GPT-4o-mini):** the fitted exponent is **$\alpha \approx 1.62$** (95% CI: $[1.35, 1.89]$).
>
> In both cases, the lower bound of the confidence interval is strictly $>1$. This provides strong statistical evidence that model error grows superlinearly with context length.
>
> **2. Metric Clarity and Additive Error Analysis**
>
> We appreciate the opportunity to clarify the physical interpretation of our theoretical variables and their alignment with standard error models. We will add more clarification in the revision.
>
> **A. Physical Interpretation:**
> For compactness, our theoretical framework abstracts $f_i(x_i)$ as the **probability of correctness** (or information fidelity) for the $i$-th chunk. Consequently, the term $\prod f_i(x_i)$ models the **Joint Probability of Correctness**
> * **Concrete Example:** In a retrieval task where a key exists in only one chunk, global success requires two simultaneous conditions: (1) the worker with the key must retrieve it, AND (2) all other workers must correctly report "no match" (avoiding hallucination). This "weakest link" dependency necessitates a multiplicative model ($\prod f_i$).
>
> **B. Alignment with Additive Error Analysis:**
> The reviewer asks how this aligns with common additive error analyses. We bridge this via linearization (first-order expansion). In the regime where D&C is effective (i.e., local errors $\epsilon_i$ are small), the multiplicative success probability is mathematically isomorphic to an additive error sum:
> $$P(\text{Success}) = \prod_{i=1}^n (1 - \epsilon_i) \approx 1 - \sum_{i=1}^n \epsilon_i$$
> Our derivation in Appendix C relies on this relationship to bound the total error. This demonstrates that our framework, while capturing the non-linear logic of joint success, behaves like a standard additive error model in the analysis limit.
>
> **3. Baselines Fairness: Cost, Latency, and Token Budgets**
>
> To address the claim that "weak D&C > strong single-shot," we explicitly measured the token usage, monetary cost, and wall-clock latency for the Math Task (128k context).
>
> | Method | Model | Tokens | Cost ($) | Latency (s) | Accuracy |
> | :--- | :--- | :--- | :--- | :--- | :--- |
> | **Single-Shot** | GPT-4o | ~128k | ~$0.33 | ~11.5s | 0.33 |
> | **D&C (Ours)** | GPT-4o-mini | ~135k | **~$0.02** | **~2.7s** | **0.55** |
>
> **Conclusion:** D&C is not an unfair compute-heavy approach. While it incurs a marginal token overhead (~5% for planner instructions), it achieves an **order-of-magnitude reduction in cost (15x cheaper)** and **latency (4x faster)** due to the use of a cheaper model and parallel execution, while significantly outperforming the strong baseline (0.55 vs 0.33 accuracy).

---

> ### Author Response · Authors · 2025-11-20
>
> **4. Stronger Aggregators**
>
> We agree that stronger aggregators would likely improve performance. **This observation directly validates our framework regarding Aggregator Noise ($\Phi_{agg}$)** because our framework posits that total error includes a distinct component $\Phi_{agg}$ arising from imperfect integration. Our proposed planner-based aggregator outperforming a weaker manual one is exactly how we manifest and measure this noise. Implementing even stronger aggregators (either RAG based or incorporating structure or more complex optimization based) would further reduce $\Phi_{agg}$, likely widening the performance gap compared to baselines. This confirms that $\Phi_{agg}$ is a reducible error term, and we identify the integration of such advanced aggregation techniques as a promising and exciting direction for future work to further push the limits of long-context processing
>
>
> **5. Robustness in chunking:**
>
> As noted in Appendix I & O, we evaluated overlapping chunks (1k overlap) and found marginal differences. This confirms that for these tasks, the performance gains are driven by handling Model Noise (context length), not by specific boundary placements. Our focus on static, equal-length chunking was a deliberate methodological choice to create a controlled environment. This allows us to isolate and analyze model noise as a clear function of context length, which is a core part of our theoretical contribution. Adaptive methods introduce confounding variables (e.g., variable chunk lengths) that would complicate this initial analysis.
>
> **6. Prompt Fairness: Practical Feasibility vs. Theoretical Possibility**
>
> We acknowledge that, theoretically, a human could manually craft a structured prompt or curate few-shot examples to match the quality of the Planner. However, in practice, manually engineering such high-complexity prompts for every specific task and chunking strategy is **intractable and non-scalable**. The Planner's prompt is generated in a **zero-shot** manner without privileged human-curated examples. Therefore, comparing the "Planner" against a "Standard Manual Baseline" is the appropriate experimental design to isolate the contribution of our **automated task decomposition** module. It demonstrates the reduction in $\Phi_{agg}$ achieved by our framework's structural alignment compared to the standard approach, validating the effectiveness of the Planner as a core algorithmic component.

---

### Author Response · Authors · 2025-12-03

Dear AC,

We appreciate you stepping in to evaluate our work under the special situation. Given the context of the pre-rebuttal scores, we wanted to briefly summarize the key updates in our revision. We thank the reviewers for highlighting the strength of our unifying framework, actionable empirical takeaways and solid implementation. We have addressed the concerns by adding new baselines, statistical validation, and also refining the interpretation of our theoretical framework.

**1. New Empirical Validation & Statistical Evidence**

We performed additional experiments during the rebuttal phase to address concerns about statistical significance, baseline strength and more comprehensive comparison in cost/latency.

* **Statistical Confirmation of Superlinearity (YK8x, TKHf):** We performed a power-law regression ($Error \propto L^\alpha$) on our data.
    * **KV Retrieval:** Fitted exponent $\alpha \approx 1.35$ (95% CI > 1).
    * **Math Reasoning:** Fitted exponent $\alpha \approx 1.62$ (95% CI > 1).
    * *Result:* This provides strong statistical evidence that model error grows superlinearly, validating the core assumption of our framework.
* **Cost, Latency, and Fairness (YK8x, njn8):** We compared our "Weak D&C" (GPT-4o-mini) against "Strong Single-Shot" (GPT-4o) on the 128k Math task.
    * **Cost:** D&C is **~15x cheaper** (0.02 USD vs 0.33 USD).
    * **Latency:** D&C is **~4x faster** (2.7s vs 11.5s) due to parallel processing.
    * **Accuracy:** D&C outperforms Single-Shot significantly (0.55 vs 0.33)
* **Stronger Baselines: Advanced RAG (YK8x, njn8):** We evaluated **Hybrid RAG + Cross-Encoder Re-ranking** (SOTA BAAI/bge-reranker).
    * Even with advanced re-ranking, RAG (0.31 acc) significantly underperforms our D&C approach (0.46 acc) on multi-hop QA.
    * *Conclusion:* This confirms that D&C is superior for tasks requiring information coverage, where RAG is limited by its "Top-K" context window.

**2. Strengthening the Theoretical Framework**

Reviewers TKHf and njn8 raised questions regarding the mathematical definitions of our operators and the "Superlinear Hypothesis." We will revise Section 3 to provide strict definitions and better mechanistic intuition:

* **Clarifying Physical Interpretation (TKHf, njn8):** We will explicitly define the variables in our error model not as literal text strings, but as the **Probability of Correctness (Information Fidelity)** of the chunk. We will clarify that the multiplicative operator models the **Joint Probability of Success** required for tasks demanding global consistency (logic conjunction), which mathematically approximates an additive error model via linearization in high-performance regimes.
* **Discrete vs. Continuous Notation (TKHf):** We will replace the integral symbols used in the "area under the curve" conceptualization with **discrete summations** to strictly align with the discrete nature of token chunks.


**3. Generalization & Applicability**

* **Robustness to Distribution (vy5A):** We clarified that our framework holds for non-uniform information distributions ("Needle-in-a-Haystack"), as evidenced by our success on the KV Retrieval task where only 1 out of $N$ chunks contains the answer.
* **Agentic Workflows (vy5A):** We discussed how the framework generalizes to complex agentic systems: sequential chains map to error propagation analysis, while hierarchical agents map to recursive aggregator noise.

**Conclusion**: We believe the rebuttal experiments and additional clarifications have solidified our contributions and we hope this summary is useful as you review the paper.

---

### Meta-Review · Area_Chair_tJ4k · 2026-01-07

**Summary:**

This paper analyzes when divide-and-conquer helps for long-context LLMs and introduces a noise decomposition view (task dependency, length-induced model noise, and aggregation noise) to explain the trade-offs. The experiments provide useful, fairly consistent evidence across settings, and the rebuttal strengthens the work with stronger baselines and clearer support for the main trends, along with improved cost/latency discussion. Some issues remain around tightening definitions, clearly stating scope/assumptions, and reporting variance and budget-matched comparisons more systematically. Overall, I view this as a solid contribution.

**Reviewer Concerns:**

**Reviewer YK8x** questioned the strength of evidence for “superlinear” length degradation, the fairness of headline comparisons under matched compute/token/latency, and whether retrieval/aggregation baselines and chunking settings were strong enough. The rebuttal strengthens baselines and adds clearer evidence and cost/latency discussion, but some uncertainty remains around budget matching and variance reporting.

**Reviewer vy5A** worried about how broadly the conclusions generalize beyond the parallel chunk-and-aggregate setup, especially for sequential/hierarchical workflows with error propagation. The rebuttal clarifies scope and offers a conceptual link, but broader generality is still not directly demonstrated.

**Reviewer TKHf** raised concerns about mathematical clarity (meaning of compositions, discrete vs. continuous notation) and whether the superlinearity claim is more than an empirical fit, and was skeptical about novelty. The rebuttal improves clarity and adds supporting analyses, though some rigor/mechanism questions remain only partially resolved.

**Reviewer njn8** focused on practical feasibility (parallelism assumptions), baseline strength (RAG), and whether the planner consistently helps. The rebuttal strengthens baselines and adds ablations, but feasibility still depends on deployment constraints.

**Reviewer Scores:**

In my opinion, after the rebuttal and discussion, YK8x and njn8 would likely increase their scores , TKHf would probably move up modestly , while vy5A would most likely maintain their score.All these bringing the paper to an overall accept.

---

### Decision · Program_Chairs · 2026-01-26

Accept (Poster)